# LEAPS: A discrete neural sampler via locally equivariant networks

**Peter Holderrieth** [* 1]   **Michael S. Albergo** [* 2 3]   **Tommi Jaakkola** [1]

## Abstract

We propose *LEAPS*, an algorithm to sample from discrete distributions known up to normalization by learning a rate matrix of a continuous-time Markov chain (CTMC). LEAPS can be seen as a continuous-time formulation of annealed importance sampling and sequential Monte Carlo methods, extended so that the variance of the importance weights is offset by the inclusion of the CTMC. To derive these importance weights, we introduce a set of Radon-Nikodym derivatives of CTMCs over their path measures. Because the computation of these weights is intractable with standard neural network parameterizations of rate matrices, we devise a new compact representation for rate matrices via what we call *locally equivariant* functions. To parameterize them, we introduce a family of locally equivariant multilayer perceptrons, attention layers, and convolutional networks, and provide an approach to make deep networks that preserve the local equivariance. This property allows us to propose a scalable training algorithm for the rate matrix such that the variance of the importance weights associated to the CTMC are minimal. We demonstrate the efficacy of LEAPS on problems in statistical physics. We provide code in https://github.com/malbergo/leaps/.

## 1. Introduction

A prevailing task across statistics and the sciences is to draw samples from a probability distribution whose probability density is known up to normalization. Solutions to this problem have applications in topics ranging across Bayesian uncertainty quantification (Gelfand & Smith, 1990), capturing

---
[*]Equal contribution  [1]Massachusetts Institute of Technology [2]Society of Fellow, Harvard University [3]Institute for Artificial Intelligence and Fundamental Interactions. Correspondence to: Peter Holderrieth <phold@mit.edu>, Michael S. Albergo <malbergo@fas.harvard.edu>.

*Proceedings of the $42^{nd}$ International Conference on Machine Learning*, Vancouver, Canada. PMLR 267, 2025. Copyright 2025 by the author(s).

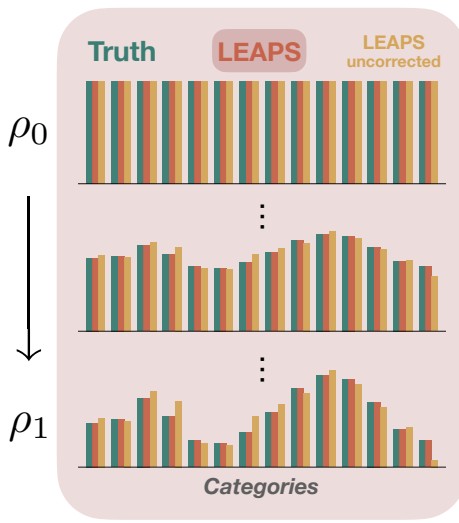

*Figure 1.* Illustration of the LEAPS algorithm. LEAPS allows to learn a dynamical transport of discrete distributions from $t = 0$ to $t = 1$ (blue). Sample are generated via the simulation of a Continuous-time Markov chain (yellow). Further, importance sampling weights allow to correct training errors trading off sample efficiency (red).

the molecular dynamics of chemical compounds (Berendsen et al., 1984; Allen & Tildesley, 2017), and computational approaches to statistical and quantum physics (Wilson, 1974; Duane et al., 1987; Faulkner & Livingstone, 2023).

The most salient approach to such sampling problems is Markov chain Monte Carlo (MCMC) (Metropolis et al., 1953; Robert et al., 1999), in which a randomized process is simulated whose equilibrium is the distribution of interest. While powerful and widely applied, MCMC methods can be inefficient as they suffer from slow convergence times into equilibrium, especially for distributions exhibiting multi-modality. Therefore, MCMC is often combined with other techniques that rely on non-equilibrium dynamics, e.g. via annealing from a simpler distribution with annealed importance sampling (AIS) (Kahn & Harris, 1951; Neal, 2001) or sequential Monte Carlo methods (SMC) (Doucet et al., 2001). Even then, the variance of these importance weights may be untenably large, and making sampling algorithms more efficient remains an active area of research. Inspired by the rapid progress in deep learning, there has been extensive

interest in augmenting contemporary sampling algorithms with learning (Noé et al., 2019; Albergo et al., 2019; Gabrié et al., 2022; Nicoli et al., 2020; Matthews et al., 2022), while still maintaining their statistical guarantees.

Recently, there has been rapid progress in development of generative models using techniques from dynamical measure transport, i.e. where data from a base distribution is transformed into samples from the target distribution via flow or diffusion processes (Ho et al., 2020; Song et al., 2020; Albergo & Vanden-Eijnden, 2022; Albergo et al., 2023; Lipman et al., 2023; Liu et al., 2022). More recently, these models could also be developed for discrete state spaces (Campbell et al., 2022; Gat et al., 2025; Shaul et al., 2024; Campbell et al., 2024) and general state spaces and Markov processes (Holderrieth et al., 2024).

While there have been various developments on adapting these non-equilibrium dynamics for sampling in *continuous* state spaces (Zhang & Chen, 2022; Vargas et al., 2023; Máté & Fleuret, 2023; Tian et al., 2024; Albergo & Vanden-Eijnden, 2024; Richter & Berner, 2024; Akhound-Sadegh et al., 2024; Sun et al., 2024), there is a lack of existing literature on such sampling approaches for discrete distributions. However, discrete data are prevalent in various applications, such as in the study of spin models in statistical physics, protein and genomic data, and language. To this end, we provide a new solution to the discrete sampling problem via CTMCs. Our method is similar in spirit to the results in (Albergo & Vanden-Eijnden, 2024; Vargas et al., 2024) but takes the necessary theoretical and computational leaps to make these approaches possible for discrete distributions. Our **main contributions** are:

- We introduce LEAPS, a non-equilibrium transport sampler for discrete distributions via CTMCs that combines annealed importance sampling and sequential Monte Carlo with learned measure transport.

- To define the importance weights, we derive a Radon-Nikodym derivative for reverse-time CTMCs, control of which minimizes the variance of these weights.

- We show that the measure transport can be learnt and the variance of the importance weights minimized by optimizing a physics-informed neural network (PINN) loss function.

- We make the computation of the PINN objective scalable by introducing the notion of a locally equivariant network. We show how to build locally equivariant versions of common neural network architectures, including attention and convolutions.

- We experimentally verify the correctness and efficacy of the resulting LEAPS algorithm in high dimensions via simulation of the Ising model.

## 2. Setup and Assumptions

In this work, we are interested in the problem of sampling from a **target distribution** $\rho_1$ on a **finite state space** $S$. We refer to $\rho_1$ by its probability mass function (pmf) given by

$$\rho_1(x) = \frac{1}{Z_1} \exp(-U_1(x)) \quad (x \in S), \tag{1}$$

where we assume that we do not know the normalization constant $Z_1$ but only the function **potential** $U_1$. Our goal is to produce samples $X \sim \rho_1$. To achieve this goal, it is common to construct a **time-dependent probability mass function (pmf)** $(\rho_t)_{0 \leq t \leq 1}$ over $S$ which fulfils that $\rho_0$ has a distribution from which we can sample easily, e.g. $\rho_0 = \text{Unif}_S$, and $\rho_1$ is our target of interest. We write $\rho_t$ as:

$$\rho_t(x) = \frac{1}{Z_t} \exp(-U_t(x)), \tag{2}$$

$$Z_t = \sum_{y \in S} \exp(-U_t(y)), \quad F_t = -\log Z_t \tag{3}$$

where $Z_t$ (or equivalently $F_t$) is unknown. The value $F_t$ is also called the **free energy**. Throughout, we assume that $U_t$ is continuously differentiable in $t$. For example, we can set $U_t(x) = tU_1(x)$ so that $\rho_0 = \text{Unif}_S$ and we get that $\rho_t \propto \exp(-tU_1(x))$ that can be considered a form of *temperature annealing*.

## 3. Sampling with CTMCs

In this work, we seek to sample from $\rho_1$ using **continuous-time Markov chains (CTMCs)**. A CTMC $(X_t)_{0 \leq t \leq 1}$ is given by a set of random variables $X_t \in S$ $(0 \leq t \leq 1)$ whose evolution is determined by a time-dependent **rate matrix** $Q_t(y, x) \in \mathbb{R}$ $(0 \leq t \leq 1, x, y \in S)$ which fulfills the conditions:

$$Q_t(y; x) \geq 0 \qquad \text{(for } y \neq x) \tag{4a}$$

$$Q_t(x; x) = -\sum_{y \neq x} Q_t(y, x) \qquad \text{(for } x \in S) \tag{4b}$$

The rate matrix $Q_t$ determines the **generator equation**

$$\mathbb{P}[X_{t+h} = y | X_t = x] = \mathbf{1}_{x=y} + hQ_t(y, x) + o(h) \tag{5}$$

for all $x, y \in S$ and $h > 0$ where $o(h)$ describes an error function such that $\lim_{h \to 0} o(h)/h = 0$. Because this equation describes the infinitesimal transition probabilities of the CTMC, we can sample from $X_t$ approximately via the following iterative Euler scheme:

$$X_{t+h} \sim \tilde{\mathbb{P}}[\cdot | X_t] := (\mathbf{1}_{X_t=y} + hQ_t(y, X_t))_{y \in S} \tag{6}$$

where $\tilde{\mathbb{P}}[\cdot | X_t]$ describes a valid probability distribution for small $h > 0$ by the conditions we imposed on $Q_t$ (see (4)).

Our goal is to find a $Q_t$ that is a solution to the **Kolmogorov forward equation (KFE)**

$$\partial_t \rho_t(x) = \sum_{y \in S} Q_t(x,y)\rho_t(y), \quad \rho_{t=0} = \rho_0. \quad (7)$$

Fulfilling the KFE is a necessary and sufficient condition to ensure that the distribution of walkers initialized as $X_0 \sim \rho_0$ and evolving according to (6) follow the prescribed path $\rho_t$, in particular such that $X_{t=1} \sim \rho_1$.

*Remark* 3.1. While the problem we focus on this work is sampling, the contributions of this work hold more generally for the problem of finding a CTMC that follows a prescribed time-varying density $\rho_t$, i.e. that the condition in (7) holds.

## 4. Proactive Importance Sampling

In general, the CTMC $(X_t)_{0 \leq t \leq 1}$ with arbitrary $Q_t$ will have different marginals than $\rho_t$. To still obtain an unbiased estimator, it is common to use **importance sampling (IS)** to reweight samples obtained while simulating $X_t$ or **sequential Monte Carlo (SMC)** (Doucet et al., 2001) to resample the walkers along the trajectory. Here, we introduce a time-evolving set of log-weights $A_t \in \mathbb{R}$ for $0 \leq t \leq 1$ to re-weight the distribution of $X_t$ to a distribution $\mu_t$ defined such that for all $h : S \to \mathbb{R}$

$$\mathbb{E}_{x \sim \mu_t}[h(x)] = \frac{\mathbb{E}[\exp(A_t)h(X_t)]}{\mathbb{E}[\exp(A_t)]}$$

$$\Leftrightarrow \quad \mu_t(x) = \frac{\mathbb{E}[\exp(A_t)|X_t = x]}{\sum_{y \in S} \mathbb{E}[\exp(A_t)|X_t = y]},$$

where $\mathbb{E}[\cdot]$ denotes expectation over the process $(X_t, A_t)$. Intuitively, the distribution $\mu_t$ is obtained by re-weighting samples from the current distribution of $X_t$. This effectively means that from a finite number of samples $(X_t^1, A_t^1), \ldots, (X_t^n, A_t^n)$, we can obtain a Monte Carlo estimator via

$$\mathbb{E}_{x \sim \mu_t}[h(x)] \approx \sum_{i=1}^{n} \frac{\exp(A_t^i)}{\sum_{j=1}^{n} \exp(A_t^j)} h(X_t^i) \quad (8)$$

Our goal is to find a scheme of computing $A_t$ such that its reweighted distribution coincides with the target densities $\rho_t$:

$$\mu_t = \rho_t \quad (0 \leq t \leq 1) \quad (9)$$

In particular, this would mean that (8) is a good approximation for large $n$.

**Proactive importance sampling.** We next propose an IS scheme of computing weights $A_t$. Before we provide a formal derivation, we provide a *heuristic* derivation of our

proposed scheme in the following paragraph. Intuitively, the log-weights $A_t$ should accumulate the deviation from the true distribution of $X_t$ to the desired distribution $\rho_t$. We can rephrase this as "accumulating the error of the KFE" that one may want to write as the difference between both sides of (7):

$$\partial_t \rho_t(x) - \sum_{y \in S} Q_t(x,y)\rho_t(y)$$

As we do not know the normalization constant $Z_t$, it is intuitive to divide by $\rho_t(x)$ to get

$$\frac{\partial_t \rho_t(x)}{\rho_t(x)} - \sum_{y \in S} Q_t(x,y)\frac{\rho_t(y)}{\rho_t(x)}$$

Using that $\partial_t \rho_t(x)/\rho_t(x) = \partial_t \log \rho_t(x) = \partial_t F_t - \partial_t U_t(x)$, we obtain after dropping the unknown time-dependent constant $\partial_t F_t$:

$$\mathcal{K}_t \rho_t(x) = -\partial_t U_t(x) - \sum_{y \in S} Q_t(x,y)\frac{\rho_t(y)}{\rho_t(x)} \quad (10)$$

where we defined a new operator $\mathcal{K}_t \rho_t$. Intuitively, the operator $\mathcal{K}_t$ measures the violation from the KFE in log-space (up to $F_t$) and it is intuitive to define $A_t$ as the accumulated error of that violation, i.e. as the integral

$$A_t = \int_0^t \mathcal{K}_s \rho_s(X_s) \mathrm{d}s \quad (11)$$

To simulate $A_t$ alongside $X_t$, we use the approximate update role:

$$A_{t+h} = A_t + h\mathcal{K}_t \rho_t(X_t)$$

We call this **proactive importance sampling (proactivate IS)** as the update operator $\mathcal{K}_t$ anticipates where $X_t$ is jumping to. We next provide a rigorous characterization of $A_t$ defined in this manner.

## 5. Proactivate IS via Radon-Nikodym Derivatives

A priori, it is not clear that the log-weights $A_t$ that we obtain via the proactive rule fulfil the desired condition in (9) (i.e. provide a valid IS scheme). Beyond showing this property, we show that there are many possible IS schemes but the proactive update rule is *optimal* among a natural family of IS schemes. To do so, we present a set of Radon-Nikodym derivatives in path space.

Let $\mathcal{X}$ be a state space and $\mathbb{P}, \mathbb{Q}$ be two probability measures over $\mathcal{X}$. Then the **Radon-Nikodym derivative (RND)** $\frac{\mathrm{d}\mathbb{Q}}{\mathrm{d}\mathbb{P}}$ allows to express expected values of $\mathbb{Q}$ via expected values

of $\mathbb{P}$. Specifically, the RND is a function $\frac{\mathrm{d}\mathbb{Q}}{\mathrm{d}\mathbb{P}} : \mathcal{X} \to \mathbb{R}$ such that for any (bounded, measurable) function $G : \mathcal{X} \to \mathbb{R}$ it holds that:

$$\mathbb{E}_{\mathbf{X} \sim \mathbb{Q}}[G(\mathbf{X})] = \mathbb{E}_{\mathbf{X} \sim \mathbb{P}}\left[G(\mathbf{X})\frac{\mathrm{d}\mathbb{Q}}{\mathrm{d}\mathbb{P}}(\mathbf{X})\right]$$

The state space that we are interested in is the space $\mathcal{X}$ of CTMC trajectories. Specifically, for a trajectory we denote with $X_{t^-} = \lim_{t' \uparrow t} X_{t'}$ the left limit and with $X_{t^+} = \lim_{t' \downarrow t} X_{t'}$ the right limit. The space $\mathcal{X}$ of CTMC trajectories is then defined as

$$\mathcal{X} = \{X : [0,1] \to S | X_{t^-} \text{ exists and } X_{t^+} = X_t\},$$

i.e. all trajectories that are continuous from the right with left limits. Such trajectories are commonly called **càdlàg** trajectories. In other words, jumps (switches between states) happen if and only if $X_{t^-} \neq X_t$. We consider *path distributions* (or *path measures*), i.e. probability distributions over trajectories. For a CTMC $\mathbf{X} = (X_t)_{0 \leq t \leq 1}$ with rate matrix $Q_t$ and initial distribution $\mu$, we denote the corresponding **path distribution** as $\overrightarrow{\mathbb{P}}^{\mu,Q}$ where the arrow $\overrightarrow{\mathbb{P}}$ denotes that we go forward in time. Similarly, we denote with $\overleftarrow{\mathbb{P}}^{\nu,Q'}$ a CTMC running in reverse time initialized with $\nu$. We present the following proposition whose proof can be found in Appendix A:

**Proposition 5.1.** *Let $\mu, \nu$ be two initial distributions over $S$. Let $Q_t, Q_t'$ be two rate matrices. Then the Radon-Nikodym derivative of the corresponding path distributions running in opposite time over the time interval $[0,t]$ is given by:*

$$\log \frac{\mathrm{d}\overleftarrow{\mathbb{P}}^{\nu,Q'}}{\mathrm{d}\overrightarrow{\mathbb{P}}^{\mu,Q}}(\mathbf{X}) = \log(\nu(X_t)) - \log(\mu(X_0))$$
$$+ \int_0^t Q_s'(X_s, X_s) - Q_s(X_s, X_s)\mathrm{d}s$$
$$+ \sum_{s, X_s^- \neq X_s} \log\left(\frac{Q_s'(X_s^-, X_s)}{Q_s(X_s, X_s^-)}\right)$$

*where we sum over all points where $X_s$ jumps in the last term.*

Let us now revisit our goal of finding an IS scheme to sample from the target distribution $\rho_1$. The key idea is to construct a CTMC running in reverse-time with initial distribution $\rho_t$ and then use the RND from Proposition 5.1. For a function $h : S \to \mathbb{R}$, we can then express its expectation under $\rho_t$ as:

$$\mathbb{E}_{x \sim \rho_t}[h(x)] = \mathbb{E}_{\mathbf{X} \sim \overleftarrow{\mathbb{P}}^{\rho_t,Q'}}[h(X_t)]$$
$$= \mathbb{E}_{\mathbf{X} \sim \overrightarrow{\mathbb{P}}^{\rho_0,Q}}\left[h(X_t)\frac{\mathrm{d}\overleftarrow{\mathbb{P}}^{\rho_t,Q'}}{\mathrm{d}\overrightarrow{\mathbb{P}}^{\rho_0,Q}}(\mathbf{X})\right] \quad (12)$$

i.e. the RND $\frac{\mathrm{d}\overleftarrow{\mathbb{P}}^{\rho_1,Q'}}{\mathrm{d}\overrightarrow{\mathbb{P}}^{\rho_0,Q}}(\mathbf{X})$ gives a valid set of importance weights. Note that this holds for *arbitrary* $Q_t'$.

However, to sample efficiently, it is crucial that the IS weights have low variance. Therefore, we will now derive the *optimal* IS scheme of this form. Ideally the weights will have *zero* variance - in other words the RND $\frac{\mathrm{d}\overleftarrow{\mathbb{P}}^{\rho_1,Q'}}{\mathrm{d}\overrightarrow{\mathbb{P}}^{\rho_0,Q}}(\mathbf{X})$ will be constant $= 1$. This is the case if and only if the path measures are the same, i.e. if the CTMC in reverse time is a time-reversal of the CTMC running in forward time. It is well-known that this is equivalent to:

$$Q_t'(y,x) = Q_t(x,y)\frac{q_t(y)}{q_t(x)} \quad \text{for all } y \neq x \quad (13)$$

where $q_t$ denotes the true marginal of $X_t$, i.e. $X_t \sim q_t$. As we strive to make $q_t = \rho_t$, it is natural to set $q_t = \rho_t$ in (13) and define $Q_t' = \bar{Q}_t$ as

$$\bar{Q}_t(y,x) = Q_t(x,y)\frac{\rho_t(y)}{\rho_t(x)} \quad \text{for all } y \neq x \quad (14a)$$

$$\bar{Q}_t(x,x) = -\sum_{y \in S, y \neq x} Q_t(x,y)\frac{\rho_t(y)}{\rho_t(x)} \quad (14b)$$

Let us now return to the proactive update that we defined in (11). We can now rigorously characterize it. Plugging in the definition of $\bar{Q}$, we can use Proposition 5.1 to obtain the main result of this section:

**Theorem 5.2.** *For the proactivate updates $A_t$ as defined in (11) and $\bar{Q}_t$ as defined in (14), it holds:*

$$A_t + F_t - F_0 = \log \frac{\mathrm{d}\overleftarrow{\mathbb{P}}^{\rho_t,\bar{Q}}}{\mathrm{d}\overrightarrow{\mathbb{P}}^{\rho_0,Q}}(\mathbf{X}) \quad (15)$$

*This implies that we obtain a valid IS scheme fulfilling:*

$$\mathbb{E}_{x \sim \rho_t}[h(x)] = \frac{\mathbb{E}[\exp(A_t)h(X_t)]}{\mathbb{E}[\exp(A_t)]} \quad (0 \leq t \leq 1) \quad (16)$$

*i.e. (9) holds. Further, $A_t$ will have zero variance for every $0 \leq t \leq 1$ if and only if $X_t \sim \rho_t$ for all $0 \leq t \leq 1$.*

A proof can be bound in Appendix B. Note that Theorem 5.2 is useful because we can, in principle, compute $A_t$, i.e. there are no unknown variables, and that this holds for arbitrary $Q_t$. This theorem can be seen as a discrete state space equivalent of the generalized version of the Jarzynski equality (Jarzynski, 1997; Vaikuntanathan & Jarzynski, 2008) that has also recently been used for sampling in continuous spaces (Vargas et al., 2024; Albergo & Vanden-Eijnden, 2024). Finally, it is important to note that the IS scheme will not have zero variance if it does not hold that $X_t \sim \rho_t$.

## 6. PINN Objective

As a next step, we introduce a learning procedure for learning an optimal rate matrix of a CTMC. For this, we denote with $Q_t^\theta$ a parameterized rate matrix with parameters $\theta$ (e.g. represented in a neural network). Our goal is to learn $Q_t^\theta$ such that $X_t \sim \rho_t$ is fulfilled for all $0 \le t \le 1$. By Theorem 5.2 this equivalent to minimizing the variance of the IS weights. To measure the variance the weights, it is common to use the log-variance divergence (Nüsken & Richter, 2023; Richter & Berner, 2023) given by

$$\mathcal{L}^{\text{log-var}}(\theta; t) = \mathbb{V}_{\mathbf{X} \sim \mathbb{Q}}[\log \frac{\mathrm{d} \overleftarrow{\mathbb{P}}^{\rho_t, \bar{Q}^\theta}}{\mathrm{d} \overrightarrow{\mathbb{P}}^{\rho_0, Q^\theta}}(\mathbf{X})]$$
$$= \mathbb{V}_{\mathbf{X} \sim \mathbb{Q}}[A_t + F_t - F_0]$$
$$= \mathbb{V}_{\mathbf{X} \sim \mathbb{Q}}[A_t]$$

where $\mathbb{Q}$ is a reference measuring whose support covers the support of $\overleftarrow{\mathbb{P}}^{\rho_t, \bar{Q}^\theta}$ and $\overrightarrow{\mathbb{P}}^{\rho_0, Q^\theta}$ and where we used that $F_0, F_t$ are constants. The above loss is tractable but we can bound it by a loss that is computationally more efficient. To do so, we use an auxiliary **free energy network** $F_t^\phi : \mathbb{R} \to \mathbb{R}$ with parameters $\phi$. Note that $F_t^\phi$ is a one-dimensional function and therefore induces minimal additional computational cost. As before, let the operator $\mathcal{K}_t^\theta$ be defined as:

$$\mathcal{K}_t^\theta \rho_t(x) = -\partial_t U_t(x) - \sum_{y \in S} Q_t^\theta(x, y) \frac{\rho_t(y)}{\rho_t(x)} \qquad (17)$$

**Proposition 6.1.** *For any reference measure $\mathbb{Q}$, the **PINN-objective** defined by*

$$\mathcal{L}(\theta, \phi; t) = \mathbb{E}_{s \sim Unif_{[0,t]}, x_s \sim \mathbb{Q}_s} \left[ |\mathcal{K}_s^\theta \rho_s(x_s) - \partial_s F_s^\phi|^2 \right]$$

*has a unique minimizer $(\theta^*, \phi^*)$ such that $Q_t^{\theta*}$ satisfies the KFE and $F_t^{\phi*} = F_t$ is the free energy. Further, this objective is an upper bound to the log-variance divergence:*

$$\mathcal{L}^{log\text{-}var}(\theta; t) \le t^2 \mathcal{L}(\theta, \phi; t)$$

*In particular, if $\mathcal{L}(\theta, \phi; t) = 0$, then also $\mathcal{L}^{log\text{-}var}(\theta; t) = 0$ and the variance of the IS weights is zero.*

A proof can be found in Appendix C. Note that we can easily minimize the PINN objective via stochastic gradient descent (see Algorithm 2). It is "off-policy" as the reference distribution $\mathbb{Q}$ is arbitrary. This objective can be seen as the CTMC equivalent of that in (Máté & Fleuret, 2023; Albergo & Vanden-Eijnden, 2024; Tian et al., 2024; Sun et al., 2024).

## 7. Adding Annealed IS and SMC

It is possible to add an arbitrary MCMC scheme to the above dynamics. This effectively combines an "unlearned"

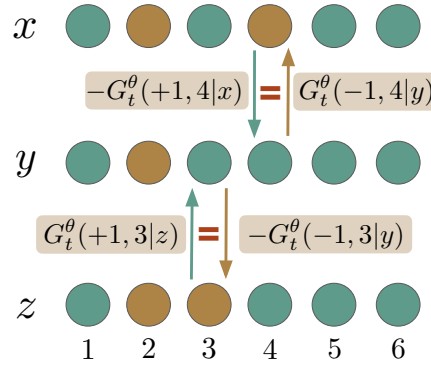

*Figure 2.* Visualization of local equivariance. Two tokens $\mathcal{T} = \{-1, +1\}$ and $d = 6$. Local equivariance means that the "flux" to transition to a neighbor is the negative of the flux of transitioning from that neighbor back.

MCMC scheme with a learned transport. Most of the time, MCMC schemes are formulated as *discrete* time Markov chains. Hence, we first describe how they can be formulated as CTMCs. For every fixed $0 \le t \le 1$, an MCMC scheme for the distribution $\rho_t$ is given by a stochastic matrix $\mathcal{M}_t(y, x)$, i.e. $\mathcal{M}_t(y, x) \ge 0$ with $\sum_{y \in S} \mathcal{M}_t(y, x) = 1$. These are constructed to satisfy the corresponding *detailed balance condition*:

$$\mathcal{M}_t(y, x) \rho_t(x) = \mathcal{M}_t(x, y) \rho_t(y) \qquad (18)$$

We can convert this into an annealed CTMC scheme by only accepting updates with a probability scaled by a parameter $\epsilon_t \ge 0$:

$$Q_t^{\text{MCMC}}(y, x) = \begin{cases} \epsilon_t \mathcal{M}_t(y, x) & \text{if } y \ne x \\ \epsilon_t (\mathcal{M}_t(x, x) - 1) & \text{if } y = x \end{cases}$$

The rate matrix $Q_t^{MCMC}$ satisfies equation 18 so that the second term of the $\mathcal{K}_t$-operator (see equation 10) vanishes:

$$\sum_{y \in S} Q_t^{\text{MCMC}}(x, y) \frac{\rho_t(y)}{\rho_t(x)} = \sum_{y \in S} Q_t^{\text{MCMC}}(y, x) = 0$$

where we used (4). This implies that the rate matrix

$$Q_t^\theta(y, x) + Q_t^{\text{MCMC}}(y, x)$$

will have the same PINN loss and the same IS weights for the same trajectories - because the $\mathcal{K}_t^\theta$ remains unchanged (Note that while the RND in (15) is the same, the path measures do change). Specifically, this means that we can sample and compute the weights via Algorithm 1. The parameter $\epsilon_t$ controls "how much local MCMC mixing" we want to induce. Note that the IS weights can be used both for reweighting at the final time or in addition to resample

**Algorithm 1** LEAPS Sampling

---

1: **Require:** $N$ time steps, $M$ walkers, model $F_t^\theta$, replay buffer $\mathcal{B}$, MCMC kernel $\mathcal{M}_t$, density $\rho_t$, coeff. $\epsilon_t \geq 0$, resample thres. $0 \leq \delta \leq 1$
2: **Init:** $X_0^m \sim \rho_0, A_0^m = 0 \quad (m = 1, \ldots, M)$
3: **Set** $h = 1/N$
4: **for** $n = 0$ to $N - 1$ **do**
5:     **for** $m = 1$ to $M$ **do**
6:         $X_t^m \sim \mathcal{M}_t(\cdot, X_t^m)$ with prob. $h\epsilon_t$ else $X_t^m$
7:         $X_{t+h}^m \sim (\mathbf{1}_{X_t=y} + hQ_t^\theta(y, X_t))_{y \in S}$
8:         $A_{t+h}^m = A_t^m + h\mathcal{K}_t^\theta \rho_t(X_t^m)$
9:     **end for**
10:    $t \leftarrow t + h$
11:    **if** $\text{ESS}(A_t) \leq \delta$ **then**
12:       $X_t = \text{resample}(X_t, A_t) \quad (m = 1, \ldots, M)$
13:       $A_t = 0 \qquad\qquad\qquad (m = 1, \ldots, M)$
14:    **end if**
15: **end for**
16: **Optional:** Store $\{(X_t^m, A_t^m, t)\}_{t,m}$ in $\mathcal{B}$.

---

the walkers along the trajectories, connecting it to the SMC literature (Doucet et al., 2001). We specify that one may want to do this whenever the effective sample size (ESS) (see Appendix G) drops below a threshold.

**Generalization of AIS and SMC.** For $Q_t^\theta = 0$, the above dynamics describe a continuous formulation of AIS that can be simulated approximately via Algorithm 1. In particular, this means that the algorithm presented here is a **strict generalization of AIS and SMC** (Neal, 2001; Doucet et al., 2001). Note that in the $h \to 0, \epsilon_t \to \infty$ limit, LEAPS would recover the exact distributions $\rho_t$ with AIS (i.e. even with $Q_t^\theta = 0$). Of course, this asymptotic limit is not realizable in practice with finite number of steps, and the inclusion of $Q_t^\theta$ allows us to sample much more efficiently while still maintaining statistical guarantees.

## 8. Efficient IS and Training via Local Equivariance

We now turn to the question of how to make the above training procedure efficient. Note that for small state spaces $S$ we could rely on analytical solutions to the KFE (Campbell et al., 2022; Shaul et al., 2024). In many applications, though, the state space $S$ is so large that we cannot store $|S|$ elements efficiently in a computer. Often state spaces $S$ are of the form $S = \mathcal{T}^d$ where $\mathcal{T} = \{1, \ldots, N\}$ is a set of $N$ tokens. We use the notation $\tau$ for a **token**, i.e. an element $\tau \in \mathcal{T}$. One then defines a notion of a **neighbor** $y$ of $x$, i.e. an element $y = (y_1, \ldots, y_d)$ that differs from $x$ in at most one dimension (i.e. $y_i \neq x_i$ for at most one $i$). We denote as $N(x)$ the set of all neighbors of $x$. We then restrict functional form of the rate matrices to only allow

for jumps to neighbors, i.e. $Q_t^\theta(y, x) = 0$ if $y \notin N(x)$. One can then use a neural network $Q_t^\theta$ represented by the function

$$Q_t^\theta : S \to (\mathbb{R}^{N-1})^d$$
$$x \mapsto (Q_t^\theta(\tau, i|x))_{i=1,\ldots,d, \tau \in \mathcal{T} \setminus \{x_i\}}$$

i.e. the neural network is given by the function $x \mapsto Q_t^\theta(\cdot|x)$ that returns for very dimension $i$ a value for every token $\tau$ different from $x_i$. We then parameterize a rate matrix via

$$Q_t^\theta(y, x) = \begin{cases} 0 & \text{if } y \notin N(x) \\ Q_t^\theta(y^j, j|x) & \text{else if } y_j \neq x_j \\ -\sum_{i,\tau} Q_t^\theta(\tau, i|x) & \text{if } x = y \end{cases} \quad (19)$$

This parameterization is commonly used in the context of discrete markov models ("discrete diffusion models") (Campbell et al., 2022; 2024). With that, the operator $\mathcal{K}_t^\theta$ in (10) becomes:

$$\mathcal{K}_t^\theta \rho_t(x) + \partial_t U_t(x)$$
$$= \sum_{\substack{i=1,\ldots,d \\ y \in N(x),\, y_i \neq x_i}} \left[ Q_t^\theta(y^i, i|x) - Q_t^\theta(x^i, i|y) \frac{\rho_t(y)}{\rho_t(x)} \right]$$

The key problem with the above update is that it requires us to evaluate the neural network $|N(x)|$ times (for every neighbor $y$). This makes computing $\mathcal{K}_t^\theta$ computationally prohibitively expensive. Hence, **with the standard rate matrix parameterization, the proactive IS sampling scheme and training via the PINN-objective is very *inefficient*.**

To make the computation of $\mathcal{K}_t^\theta$ efficient, we choose to induce an **inductive bias** into our neural network architecture to compute $\mathcal{K}_t^\theta$ with no additional cost. Specifically, we introduce here the notion of **local equivariance**. A neural network $G_t^\theta$ represented by the function

$$G_t^\theta : S \to (\mathbb{R}^{N-1})^d$$
$$x \mapsto (G_t^\theta(\tau, i|x))_{i=1,\ldots,d, \tau \in \mathcal{T} \setminus \{x_i\}}$$

is called **locally equivariant** if the following condition holds:

$$G_t^\theta(\tau, i|x) = -G_t^\theta(x^i, i|\text{Swap}(x, i, \tau)) \quad (i = 1, \ldots, d)$$
$$\text{where} \quad \text{Swap}(x, i, \tau) = (x_1, \ldots, x_{i-1}, \tau, x_{i+1}, \ldots, x_d)$$

In other words, the function $G_t^\theta$ gives the "flux of probability" going from $x$ to each neighbor. Local equivariance says that the flux from $x$ to its neighbor is negative the flux from the neighbor to $x$ (see Figure 2).

Therefore, every coordinate map $F_j$ is equivariant with respect to transformations of the $j$-th input ("locally" equivariant). Note that we do not specify how $F_i$ transforms for

$i \neq j$ under transformations of $x_j$. This distinguishes it from "full" equivariance as, for example, used in geometric deep learning (Bronstein et al., 2021; Weiler & Cesa, 2019; Thomas et al., 2018). We can use a locally equivariant neural network to parameterize a rate matrix via:

$$Q_t^\theta(\tau, j|x) = [G_t^\theta(\tau, j|x)]_+ \tag{20}$$

where $[z]_+ = \max(z, 0)$ describes the ReLU operation. This representation is not a restriction (see Appendix D for a proof):

**Proposition 8.1** (Universal representation theorem)**.** *For any CTMC as in (19) with marginals $\rho_t$, there is a corresponding CTMC with the same marginals $\rho_t$ and a rate matrix that can be written as in (20) for a locally equivariant function $G_t^\theta$.*

Note that this representation implies that rates only go from $x$ to $y$ or from $y$ to $x$, and in that sense are kinetically favorable (Shaul et al., 2024). Crucially, this representation allows to efficiently compute $\mathcal{K}_t^\theta$ in one forward pass of the neural network:

$$\mathcal{K}_t^\theta \rho_t(x) + \partial_t U_t(x)$$
$$= \sum_{\substack{i=1,\ldots,d \\ y \in N(x), y_i \neq x_i}} \left[ [G_t^\theta(y^i, i|x)]_+ - [-G_t^\theta(y^i, i|x)]_+ \frac{\rho_t(y)}{\rho_t(x)} \right]$$

With this, we can efficiently compute the proactive IS update $A_t$ and evaluate the PINN-objective. Therefore, this construction allows for scalable training and efficient proactivate importance sampling. We call the resulting algorithm **LEAPS** (**L**ocally **E**quivariant discrete **A**nnealed **P**roactivate **S**ampler). The acronym also highlights that we use a Markov *jump* process to sample (i.e. that takes "leaps" through space).

*Remark* 8.2. It is important to note that with the above construction, we would need to naively evaluate $\rho_t(x)$ as often as $d$ times for a single computation of $\mathcal{K}_t^\theta$. However, note that the sum goes over all neighbors over $x$. Therefore, this can be a considered as computing a *discrete gradient*. Such ratios can often be computed efficiently, e.g. for many scientific and physical models it is often only $2\times$ the computation compared to a single evaluation of $\rho_t(x)$.

## 9. Design of Locally Equivariant Networks

It remains to be stated how to construct locally equivariant neural networks - a question we turn to in this section. We will focus on three fundamental designs used throughout deep learning: Multilayer perceptrons (MLPs), attention layers, and convolutional neural networks. Usually, tokens are embedded as token vectors $e_\tau \in \mathbb{R}^{c_{in}}$ where $c_{in}$ is the embedding dimension. We therefore consider the embedded

sequence of vectors: $x = (x_1, \ldots, x_d) \in (\mathbb{R}^{c_{in}})^d$ as the input to the neural network. The specific designs we discuss here are based on two components: (1) a locally-invariant **prediction head** given by

$$H_t^\theta : (\mathbb{R}^{c_{in}})^d \to (\mathbb{R}^{c_{out}})^d$$
$$x \mapsto (H_t^\theta(1|x), \ldots, H_t^\theta(d|x))$$

i.e. $H_t^\theta$ fulfills that $H_t^\theta(i|\text{Swap}(x, i, \tau)) = H_t^\theta(i|x)$ for all $x, i, \tau$. (2) This is combined with a small network $P_t^\theta : \mathcal{T} \to \mathbb{R}^k$ that we call **token projector**. The full network is then given by

$$G_t^\theta(\tau, j|x) = (P_t^\theta(e_\tau) - P_t^\theta(x_j))^T H_t^\theta(j|x)$$

In Appendix E, we verify that $G_t^\theta$ defined in this way is locally equivariant. We discuss specific instances of this design.

**Multilayer perceptron (MLP).** Let us set $c_{in} = 1$ in this paragraph for readability. Let $W^1, \ldots, W^k \in \mathbb{R}^{d \times d}$ be a set of weight matrices with a zero diagonal, i.e. $W_{ii} = 0$ for $i = 1, \ldots, d$. Further, let $\sigma : \mathbb{R} \to \mathbb{R}$ be an activation function and $\omega_\tau \in \mathbb{R}^k$ be a learnable projection vector for every token $\tau \in \mathcal{T}$. Then define the map:

$$G_t^\theta(\tau, j|x) = \sum_{i=1}^k (\omega_\tau^i - \omega_{x_j}^i) \sigma(W^i x)_j$$

where $\sigma(W^i x)_j$ denotes the $j$-th element of the vector obtained by multiplying the vector $x$ with the matrix $W^i$ and applying the activation function $a$ componentwise. This is a locally equivariant MLP with one hidden layer.

**Locally-equivariant attention (LEA) layer.** Let us consider a self-attention layer operating on keys $k_j = k_j(x_j)$, queries $q_j = q_j(x_j)$, and values $v_j = v_j(x_j)$ - each of which is a function of element $x_j$. We define the locally equivariant attention layer then as:

$$G_t^\theta(\tau, j|x) = (\omega_\tau - \omega_{x_j})^T \left[ \sum_{s \neq j} \frac{\exp(k_s^T q_j)}{\sum_{t \neq j} \exp(k_t^T q_j)} v_s \right]$$

It can be shown that this layer is locally equivariant if the queries $q_j$ are independent of the sign of $x_j$ (i.e. $q_j(x_j) = q_j(-x_j)$) which can be easily achieved. By stacking across multiple attention heads, one can create a locally equivariant MultiHeadAttention (LEA) with this.

**Hierchical local equivariance.** Local equivariance is different from "proper" equivariance in that **the composition of locally equivariant functions is not locally equivariant** in general. Therefore, we cannot simply compose locally equivariant neural network layers as we would do

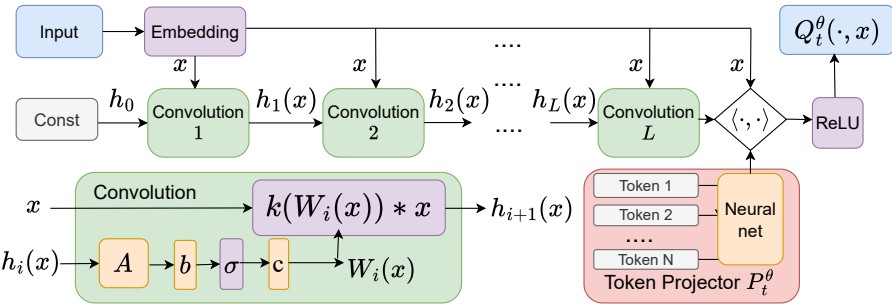

*Figure 3.* Overview of locally equivariant convolutional neural network architecture.

with "proper" equivariant neural networks. In particular, the above MLP and the attention layers cannot simply be composed as their composition would violate the local equivariance. This fundamentally changes considerations about how to compose layers and how to construct *deep* neural networks. We will now illustrate this for the case of convolutional neural networks.

**Locally-equivariant convolutional (LEC) network.** To construct a locally equivariant convolutional neural network (LEC), we assume that our data lies on a grid. A convolutional layer is characterized by a matrix $W \in \mathbb{R}^{(2k-1) \times (2k-1)}$ and its operation is denoted via $k(W) * x$ where $k$ denotes the convolutional kernel with weights $W$. Here, we set the center of $W$ to zero: $W_{kk} = 0$ (i.e. such that corresponding location is effectively ignored). To stack such layers, we can make the output of the previous layer feed into the *weights* of the next layer:

$$W_i = \sigma(A_i h_i + b_i) + c_j \qquad (i = 1, \dots, L)$$
$$h_{i+1} = k_t(W_i) * x \qquad (i = 1, \dots, L)$$
$$H_t^\theta(x) = h_L$$

where $A_i \in \mathbb{R}^{d_i \times d_i}, b_i \in \mathbb{R}^{d_i}, c_i \in \mathbb{R}^{d_i}$ are learnable tensors which operate on each coordinate independently (i.e. a 1x1 convolution) and $\sigma : \mathbb{R} \to \mathbb{R}$ is an activation function to make it non-linear. With this construction, one can stack deep highly complex convolutional neural networks. Note that this convolutional neural network has two (separate) symmetries: it is geometrically translation equivariant and locally equivariant in the sense defined in this work.

## 10. Related Work

**CTMCs.** CTMCs (Campbell et al., 2022) have been used for various applications in generative modeling ("discrete diffusion" models), including text and image generation (Shi et al., 2024; Gat et al., 2025; Shaul et al., 2024; Sahoo et al., 2024) and molecular design (Gruver et al., 2023; Campbell et al., 2024; Lisanza et al., 2024). While here we use a RND for CTMCs running in *reverse* time, one recovers the loss functions of these generative models considering a RND of two *forward* time CTMCs (see Appendix A).

**Transport and sampling.** Over the past decade there has been continued interest in combining MCMC and IS with learning transport maps. A non-parametric version of this is described in (Marzouk et al., 2016), and a parametric version through coupling-based normalizing flows was used to study systems in molecular dynamics and statistical physics (Noé et al., 2019; Albergo et al., 2019; Gabrié et al., 2022; Wang et al., 2022). These methods were extended to weave normalizing flows with SMC moves (Arbel et al., 2021; Matthews et al., 2022). More recent research focuses on replacing the generative model with a continuous flow or diffusion (Zhang & Chen, 2022; Vargas et al., 2023; Akhound-Sadegh et al., 2024; Sun et al., 2024). Our method is inspired by approaches combining measure transport with MCMC schemes (Albergo & Vanden-Eijnden, 2024; Vargas et al., 2024) and other samplers relying on PINN-based objectives in continuous spaces (Máté & Fleuret, 2023; Tian et al., 2024; Sun et al., 2024).

**Discrete Neural samplers.** The primary alternative to what we propose is to correct using importance weights arising from the estimate of the probability density computed using an autoregressive model (Nicoli et al., 2020; Wu et al., 2019; McNaughton et al., 2020). However, the computational cost of producing samples in this case scales naively as $O(d)$, whereas we have no such constraint *a priori* in our case so long as the error in the Euler sampling scheme is kept small. Other work focuses on discrete formulations of normalizing flows, but the performant version reduces to an autoregressive model (Tran et al., 2019). Recent work has considered using CTMCs for sampling by parameterizing their evolution operators directly via tensor networks (Causer et al., 2025) as opposed to neural network representations of rate matrices here. Finally, discrete diffusion models have already been explored in the context of unsupervised neural optimization (Sanokowski et al., 2024; 2025).

## 11. Experiments

As a demonstration of the validity of LEAPS in high dimensions, we benchmark it on models of statistical physics, namely the Ising model and the Potts model. We apply

LEAPS on each model on a $15 \times 15$ lattice corresponding to a $d = 15 \times 15 = 225$ dimensional space. To construct $\rho_t$, we use temperature annealing making the inverse temperature $\beta_t$ a linear function of time.

**Ablation - Ising model.** To establish the validity of our method also experimentally, we compare our results against a ground truth of long-run Glauber dynamics, an efficient algorithm for simulation in this parameter regime. In Figure 8, one can see that LEAPS recovers the expected physical observables. Further, in Figure 8 we compare three different realizations of our method, one using LEA, and the other two using deep LEC that vary in depth. The deep LEC network performs best and we use it for all subsequent experiments.

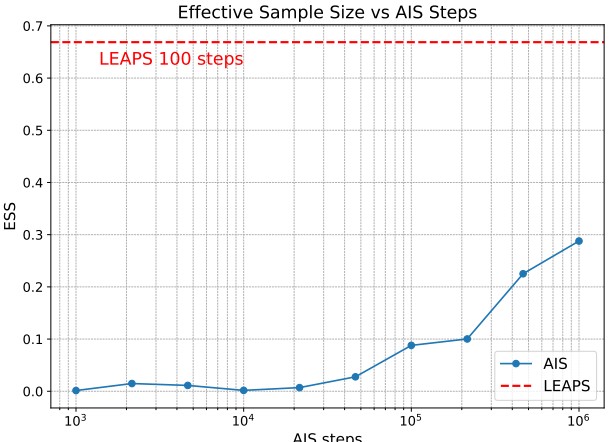

*Figure 4.* LEAPS vs AIS on Ising model. Comparison across different number of simulation steps for AIS. LEAPS is run for 100 steps. Note, however, that LEAPS requires neural net evaluations.

**Benchmarks - Ising model.** We then apply LEAPS on the parameters corresponding to the corresponding critical temperatures of the Ising model, i.e. the "hardest" temperature to sample from. Here, LEAPS achieves a high effective sample size (ESS) of $\sim 68\%$. In Figure 4, we compare LEAPS against annealed importance sampling (AIS) and see that AIS needs around $10^6$ integration steps to achieve a similar ESS (vs. 100 integration steps from LEAPS). Next, we use the DISCS benchmark (Goshvadi et al., 2023) to benchmark LEAPS against MCMC samplers (see Figure 5). The ESS of LEAPS is significantly higher, even after accounting for the number of function evaluations (NFEs).

**Benchmarks - Potts model.** We also apply the LEAPS method on the Potts model following Goshvadi et al. (2023). LEAPS achieves a high ESS ($\sim 20\%$) in only 100 integration steps while AIS has a lower ESS even for $10^6$ integration steps (see Figure 6). Also on the DISCS benchmark, LEAPS shows a higher ESS than MCMC samplers.

Overall, the above results demonstrate the validity of LEAPS and that post-training, it is able to generate realistic samples with few simulation steps, while leveraging

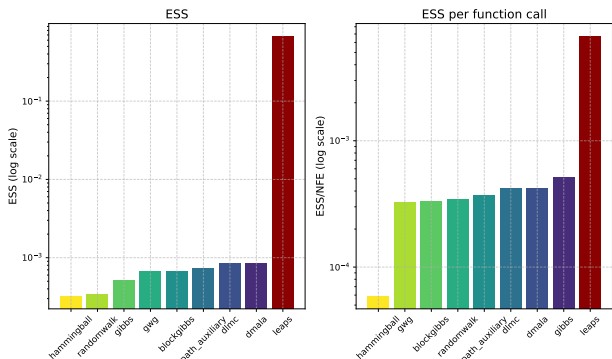

*Figure 5.* LEAPS vs MCMC samplers on Ising model (DISCS benchmark (Goshvadi et al., 2023)). Note that the comparison has limitations: Function call for LEAPS is neural network evaluations and energy calls for MCMC samplers. Further, ESS is measured differently for MCMC samplers.

importance weights to recover exact observables.

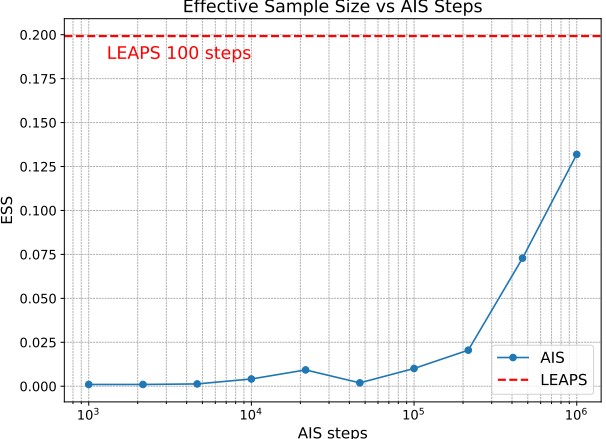

*Figure 6.* LEAPS vs AIS on Potts model for critical temperature. Comparison across different number of simulation steps for AIS. LEAPS is run for 100 steps.

## 12. Discussion

In this work, we present the LEAPS algorithm to learn to sample from discrete distributions via continuous-time Markov chains ("discrete diffusion"). Crucially, we translated the sampling problem into a neural network design problem. While we presented first locally equivariant neural network designs, we anticipate that future work will focus on building more scalable and more expressive locally equivariant network architectures. Another direction of future work will be to connect the ideas presented here with guidance and reward fine-tuning of generative model - a problem strongly tied to sampling. Further, neural samplers like LEAPS can be used to sample across a whole family of distributions as opposed to only for a single target.

## Impact Statement

This paper presents work whose goal is to advance the field of Machine Learning. There are many potential societal consequences of our work, none which we feel must be specifically highlighted here.

## Acknowledgements

We thank Eric Vanden-Eijnden for useful discussions. PH acknowledges support from the Machine Learning for Pharmaceutical Discovery and Synthesis (MLPDS) consortium, the DTRA Discovery of Medical Countermeasures Against New and Emerging (DOMANE) threats program, and the NSF Expeditions grant (award 1918839) Understanding the World Through Code. MSA is supported by a Junior Fellowship at the Harvard Society of Fellows as well as the National Science Foundation under Cooperative Agreement PHY-2019786 (The NSF AI Institute for Artificial Intelligence and Fundamental Interactions, http://iaifi.org/).

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

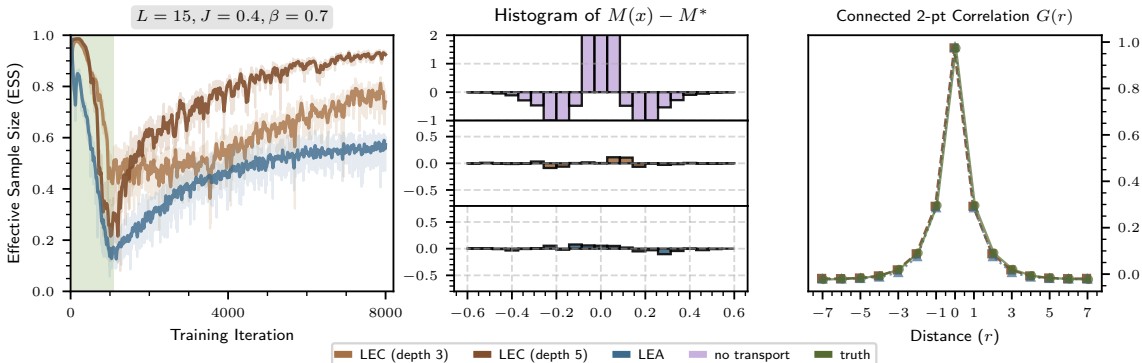

*Figure 7.* Ablation experiments on simple Ising model with $L = 15, J = 0.4, \beta = 0.7$ with the LEA and LEC architectures. **Left:** Effective sample size of LEAPS samplers over training. Green area denotes an annealing phase during training where $t$ is increased from 0 to 1. Increasing the depth of LEC significantly improves performance. For the locally equivariant attention (LEA) mechanism, we use 40 attention heads, each with query, key, and value matrices of dimension 40x40. As such, there are about 350,000 parameters in the model. In addition, the locally equivariant convolutional net (LEC) of depth three uses kernel sizes of [5, 7, 15], while the depth five version uses [3, 5, 7, 9, 15], amounting to around 100,000 parameters. **Center:** Difference in the histograms of the magnetization $M(x)$ of configurations as compared to the ground truth set attained from a Glauber dynamics run of 25,000 steps, labeled as $M^*$. We denote by "no transport" the case of using annealed dynamics with just the marginal preserving MCMC updates to show that the transport from $Q_t$ is essential in our construction. **Right:** Comparison of the 2-point correlation function for the LEA and LEC samplers against the Glauber dynamics ground truth.

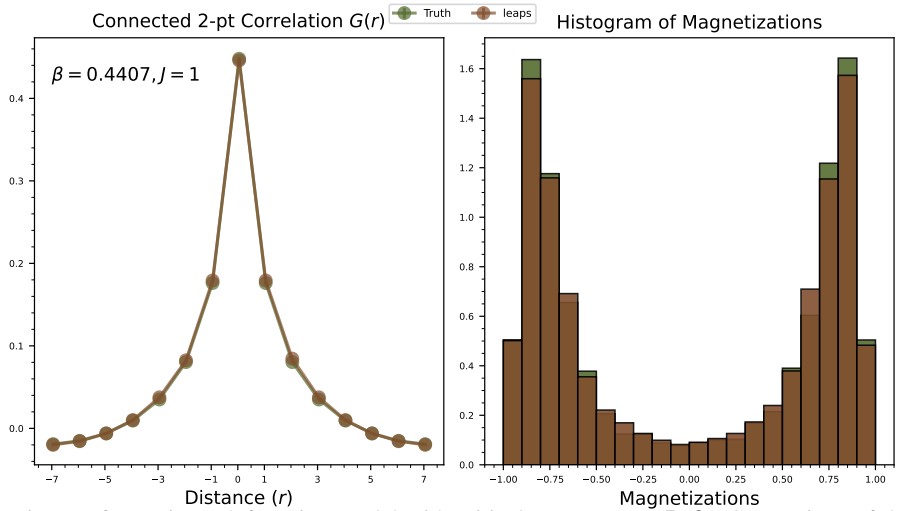

*Figure 8.* Repeat experiments from Figure 8 for Ising model with critical temperature. **Left:** Comparison of the 2-point correlation function against the Glauber dynamics ground truth. **Right:** Histograms of the magnetization $M(x)$ of configurations as compared to the ground truth set attained from a Glauber dynamics run of 25,000 steps. LEAPS recovers the expected observables.

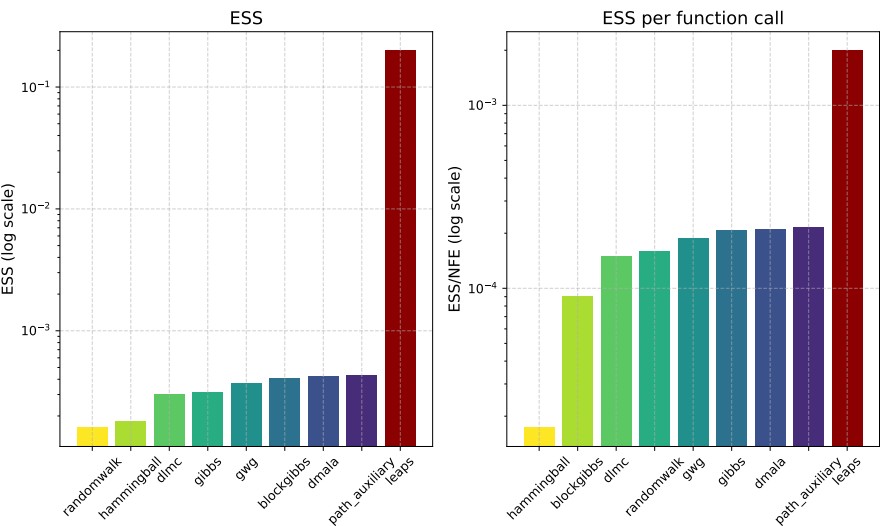

*Figure 9.* LEAPS vs MCMC samplers on Potts model (DISCS benchmark (Goshvadi et al., 2023)). Note that the comparison has limitations: Function call for LEAPS is neural network evaluations and energy calls for MCMC samplers. Further, ESS is measured differently for MCMC samplers.

---

**Algorithm 2** LEAPS training with optional replay buffer

---

**Require:** $B$ batch size, $N$ time steps, model $G_t^\theta$, free energy net $F_t^\phi$, learning rate $\eta$, replay buffer $\mathcal{B}$.
1: **while** not converged **do**
2:     **if** `use_buffer` **then**
3:         $(X_{t_m}^m, A_{t_m}^m, t_m)_{m=1,\dots,B} \leftarrow \text{SampleBatch}(\mathcal{B})$
4:     **else**
5:         $(X_{t_m}^m, A_{t_m}^m, t_m)_{m=1,\dots,B} \leftarrow \text{Algorithm 1}$
6:     **end if**
7:     $\mathcal{L}(\theta, \phi) = \frac{1}{B} \sum_m |\mathcal{K}_{t_m}^\theta \rho_{t_m}(X_{t_m}^m) - \partial_{t_m} F_{t_m}^\phi|^2$
8:     $\theta \leftarrow \theta - \eta \nabla_\theta \mathcal{L}(\theta, \phi)$
9:     $\phi \leftarrow \phi - \eta \nabla_\phi \mathcal{L}(\theta, \phi)$
10: **end while**

---

## A. Proof of Proposition 5.1

Without loss of generality, we set the final time point to be $t = 1$. We compute for a bounded continuous function $\Phi : \mathcal{X} \to \mathbb{R}$:

$$
\begin{aligned}
&\mathbb{E}_{\mathbf{X} \sim \overrightarrow{\mathbb{P}}^{\mu,Q}}[\Phi(\mathbf{X})] \\
&= \lim_{n \to \infty} \mathbb{E}_{\mathbf{X} \sim \overrightarrow{\mathbb{P}}^{\mu,Q}}[\Phi(X_0, X_{1/n}, X_{2/n}, \ldots, X_{\frac{n-1}{n}}, X_1)] \\
&= \lim_{n \to \infty} \mathbb{E}_{\mathbf{X} \sim \overleftarrow{\mathbb{P}}^{\nu,Q'}}\left[\Phi(X_0, X_{1/n}, X_{2/n}, \ldots, X_{\frac{n-1}{n}}, X_1)\frac{\overrightarrow{\mathbb{P}}^{\mu,Q}(X_0, X_{1/n}, \ldots, X_{\frac{n-1}{n}}, X_1)}{\overleftarrow{\mathbb{P}}^{\nu,Q'}(X_0, X_{1/n}, X_{2/n}, \ldots, X_{\frac{n-1}{n}}, X_1)}\right] \\
&= \lim_{n \to \infty} \mathbb{E}_{\mathbf{X} \sim \overleftarrow{\mathbb{P}}^{\nu,Q'}}\left[\Phi(X_0, X_{1/n}, X_{2/n}, \ldots, X_{\frac{n-1}{n}}, X_1)\frac{\mu(X_0)}{\nu(X_1)}\prod_{s=0,1/n,2/n,\ldots,\frac{n-1}{n}}\frac{\overrightarrow{\mathbb{P}}^{\mu,Q}(X_{s+h}|X_s)}{\overleftarrow{\mathbb{P}}^{\nu,Q'}(X_s|X_{s+h})}\right] \\
&= \lim_{n \to \infty} \mathbb{E}_{\mathbf{X} \sim \overleftarrow{\mathbb{P}}^{\nu,Q'}}\left[\Phi(\mathbf{X})\frac{\mu(X_0)}{\nu(X_1)}\exp\left(h\sum_{s,X_{s+h}=X_s}Q_s(X_s, X_s) - Q'_{s+h}(X_s, X_s)\right)\prod_{s,X_{s+h}\neq X_s}\frac{Q_s(X_{s+h}, X_s)}{Q'_{s+h}(X_s, X_{s+h})}\right] \\
&= \mathbb{E}_{\mathbf{X} \sim \overleftarrow{\mathbb{P}}^{\nu,Q'}}\left[\Phi(\mathbf{X})\frac{\mu(X_0)}{\nu(X_1)}\exp\left(\int_0^1 Q_s(X_s, X_s) - Q'_s(X_s, X_s)\mathrm{d}s\right)\prod_{s,X_{s^-}\neq X_s}\frac{Q_s(X_s, X_{s^-})}{Q'_s(X_{s^-}, X_s)}\right]
\end{aligned}
$$

where we used the definition of the rate matrix $Q_t, Q'_t$, the continuity of $Q'_t$ in $t$ and the fact that the left and right Riemann integral coincide. As $\Phi$ was arbitrary, the RND is given by:

$$
\log\frac{\mathrm{d}\overrightarrow{\mathbb{P}}^{\mu,Q}}{\mathrm{d}\overleftarrow{\mathbb{P}}^{\nu,Q'}}(\mathbf{X}) = \log(\mu(X_0)) - \log(\nu(X_1)) + \int_0^1 Q_s(X_s, X_s) - Q'_s(X_s, X_s)\mathrm{d}s + \sum_{s,X_s^-\neq X_s}\log\left(\frac{Q_s(X_s, X_s^-)}{Q'_s(X_s^-, X_s)}\right)
$$

## B. Proof of Theorem 5.2

Specifically, we use Proposition 5.1 to compute

$$
\begin{aligned}
\log\frac{\mathrm{d}\overleftarrow{\mathbb{P}}^{\rho_t,\bar{Q}_t}}{\mathrm{d}\overrightarrow{\mathbb{P}}^{\rho_0,Q_t}}(\mathbf{X}) &= \log(\rho_t(X_t)) - \log(\rho_0(X_0)) + \int_0^t \bar{Q}_s(X_s, X_s) - Q_s(X_s, X_s)\mathrm{d}s + \sum_{s,X_s^-\neq X_s}\log\left(\frac{\bar{Q}_s(X_s^-, X_s)}{Q_s(X_s, X_s^-)}\right) \\
&= F_t - F_0 - U_t(X_t) + U_0(X_0) + \int_0^t \bar{Q}_s(X_s, X_s) - Q_s(X_s, X_s)\mathrm{d}s + \sum_{s,X_s^-\neq X_s}\log\left(\frac{\bar{Q}_s(X_s^-, X_s)}{Q_s(X_s, X_s^-)}\right)
\end{aligned}
$$

Note that the function $t \mapsto U_t(X_t)$ is a piecewise differentiable function. Therefore, we can apply the fundamental theorem on every differentiable "piece" and get:

$$
\begin{aligned}
U_t(X_t) - U_0(X_0) &= \int_0^t \partial_s U_t(X_t)\mathrm{d}s + \sum_{s,X_s^-\neq X_s}U_s(X_s) - U_s(X_s^-) \\
&= \int_0^t \partial_s U_s(X_s)\mathrm{d}s + \sum_{s,X_s^-\neq X_s}\log\frac{\rho_s(X_s^-)}{\rho_s(X_s)}
\end{aligned}
$$

Next, we can insert the above equation and get:

$$\log \frac{\mathrm{d}\overleftarrow{\mathbb{P}}^{\rho_t,\bar{Q}_t}}{\mathrm{d}\overrightarrow{\mathbb{P}}^{\rho_0,Q_t}}(\mathbf{X})$$

$$=F_t - F_0 - U_t(X_t) + U_0(Y_0) + \int_0^t \bar{Q}_s(X_s, X_s) - Q_s(X_s, X_s)\mathrm{d}s + \sum_{s,X_s^- \neq X_s} \log\left(\frac{\bar{Q}_s(X_s^-, X_s)}{Q_s(X_s, X_s^-)}\right)$$

$$=F_t - F_0 - \int_0^t \partial_s U_s(X_s)\mathrm{d}s - \sum_{s,X_s^- \neq X_s} \log \frac{\rho_s(X_s^-)}{\rho_s(X_s)} + \int_0^t \bar{Q}_s(X_s, X_s) - Q_s(X_s, X_s)\mathrm{d}s + \sum_{s,X_s^- \neq X_s} \log\left(\frac{\bar{Q}_s(X_s^-, X_s)}{Q_s(X_s, X_s^-)}\right)$$

$$=F_t - F_0 - \int_0^t \partial_s U_s(X_s)\mathrm{d}s + \int_0^t \bar{Q}_s(X_s, X_s) - Q_s(X_s, X_s)\mathrm{d}s + \sum_{s,X_s^- \neq X_s} \log\left(\underbrace{\frac{\bar{Q}_s(X_s^-, X_s)}{Q_s(X_s, X_s^-)}\frac{\rho_s(X_s)}{\rho_s(X_s^-)}}_{=1}\right)$$

$$=F_t - F_0 - \int_0^t \partial_s U_s(X_s)\mathrm{d}s + \int_0^t -\sum_{y \neq X_s} Q_s(X_s, y)\frac{\rho_t(y)}{\rho_t(X_s)} - Q_s(X_s, X_s)\mathrm{d}s + 0$$

$$=F_t - F_0 + \left[-\int_0^t \partial_s U_s(X_s)\mathrm{d}s - \int_0^t \sum_{y \in S} Q_s(X_s, y)\frac{\rho_t(y)}{\rho_t(X_s)}\mathrm{d}s\right]$$

$$=F_t - F_0 + A_t$$

where we used the definition of $A_t$ in (11) and the definition of $\bar{Q}_t$ in (14). Note that for $h = 1$, we get that

$$1 = \mathbb{E}_{x \sim \rho_t}[h(x)] = \mathbb{E}[\exp(A_t + F_t - F_0)] = \mathbb{E}[\exp(A_t)]\exp(F_t - F_0)$$

because $F_t, F_0$ are constants. Therefore, in particular $\mathbb{E}[\exp(A_t)] = \exp(F_0 - F_t) = Z_t/Z_0$. Note that we assume that $Z_0 = 1$ as we know $\rho_0$. Therefore, $\mathbb{E}[\exp(A_t)] = Z_t$. This proves (16).

## C. Proof of Proposition 6.1

We can use the variational formulation of the variance as the minimizer of the mean squared error to derive a computationally more efficient upper bound, i.e. we can re-express for every $0 \leq t \leq 1$:

$$\mathcal{L}^{\text{log-var}}(\theta; t)$$

$$=\mathbb{V}_{\mathbf{X} \sim \mathbb{Q}}[A_t]$$

$$= \min_{\hat{F}_t \in \mathbb{R}} \mathbb{E}_{\mathbf{X} \sim \mathbb{Q}}[|A_t - \hat{F}_t|^2]$$

$$=t^2 \min_{\partial_s \hat{F}_s \in \mathbb{R}, 0 \leq s \leq t} \mathbb{E}_{\mathbf{X} \sim \mathbb{Q}}\left[|\frac{1}{t}\int_0^t \mathcal{K}_s^\theta \rho_s(X_s) - \partial_s \hat{F}_s \mathrm{d}s|^2\right]$$

$$\leq t^2 \min_{\partial_s \hat{F}_s \in \mathbb{R}, 0 \leq s \leq t} \mathbb{E}_{\mathbf{X} \sim \mathbb{Q}}\left[\frac{1}{t}\int_0^t |\mathcal{K}_s^\theta \rho_s(X_s) - \partial_s \hat{F}_s|^2 \mathrm{d}s\right]$$

$$=t^2 \min_{\partial_s \hat{F}_s \in \mathbb{R}, 0 \leq s \leq t} \mathbb{E}_{s \sim \text{Unif}_{[0,1]}, X_s \sim \mathbb{Q}_s}\left[|\mathcal{K}_s^\theta \rho_s(X_s) - \partial_s \hat{F}_s|^2\right]$$

where we used Jensen's inequality and denote with $\mathbb{Q}_s$ the marginal of $\mathbb{Q}$ at time $s$. We now arrive at the result by replacing the above with the free energy network $F_t^\phi$. Further, note that the above bound is tight for $\mathbb{Q}$-almost every $\mathbf{X}$:

$$\mathcal{K}_s^\theta \rho_s(X_s) - \partial_s F_s = C(\mathbf{X}_{0:t})$$

is a constant in time $s$. However, this constant may depend on $\mathbf{X}$.

# D. Proof of Proposition 8.1

Before we prove the statement, we prove an auxiliary statement about one-way rate matrices. We call a rate matrix $Q_t$ a **one-way** rate matrix if

$$Q_t(y,x) \neq 0 \quad \Rightarrow Q_t(x,y) = 0 \quad \text{for all } x \neq y$$
$$\Leftrightarrow \quad Q_t(y,x) = 0 \quad \text{or} \quad Q_t(x,y) = 0 \quad \text{for all } x \neq y$$

Intuitively, a rate matrix $Q_t$ is a one-way rate matrix if we can always only go from $x \to y$ or from $y \to x$. The next proposition shows that there is no problem restricting ourselves to one-way rate matrices.

**Lemma D.1.** *For every CTMC with rate matrix $Q_t$ and marginals $q_t$, there is a one-way rate matrix $\bar{Q}_t$ such that its corresponding CTMC $X_t$ has marginals $q_t$ if $X_0 \sim q_0$ is initialized with the same initial distribution. Furhter, if $Q_t(y,x) = 0$ for $y \neq x$, then also $\bar{Q}_t(y,x) = 0$.*

*Proof.* Let $Q_t$ be a rate matrix defining a CTMC with marginals $q_t$. Then

$$\partial_t q_t(x) = \sum_{y \in S} Q_t(x,y) q_t(y)$$

$$= \sum_{y \neq x} Q_t(x,y) q_t(y) - Q_t(y,x) q_t(x)$$

$$= \sum_{y \neq x} \left[ Q_t(x,y) - Q_t(y,x) \frac{q_t(x)}{q_t(y)} \right] q_t(y)$$

$$= \sum_{y \neq x} \left[ Q_t(x,y) - Q_t(y,x) \frac{q_t(x)}{q_t(y)} \right]_+ q_t(y) - \left[ Q_t(y,x) \frac{q_t(x)}{q_t(y)} - Q_t(x,y) \right]_+ q_t(y)$$

$$= \sum_{y \neq x} \left[ Q_t(x,y) - Q_t(y,x) \frac{q_t(x)}{q_t(y)} \right]_+ q_t(y) - \left[ Q_t(y,x) - Q_t(x,y) \frac{q_t(y)}{q_t(x)} \right]_+ q_t(x)$$

$$= \sum_{y \neq x} \bar{Q}_t(x,y) q_t(y) - \bar{Q}_t(y,x) q_t(x)$$

$$= \sum_{y \in S} \bar{Q}_t(x,y) q_t(y)$$

where we defined

$$\bar{Q}_t(y,x) = \begin{cases} \left[ Q_t(y,x) - Q_t(x,y) \frac{q_t(y)}{q_t(x)} \right]_+ & y \neq x \\ -\sum_{z \neq x} Q_t(z,x) & y = x \end{cases}$$

Note that

$$\bar{Q}_t(y,x) > 0$$

$$\Rightarrow \quad Q_t(y,x) > Q_t(x,y) \frac{q_t(y)}{q_t(x)}$$

$$\Rightarrow \quad Q_t(y,x) \frac{q_t(x)}{q_t(y)} > Q_t(x,y)$$

$$\Rightarrow \quad \left[ Q_t(x,y) - Q_t(y,x) \frac{q_t(x)}{q_t(y)} \right]_+ = 0$$

$$\Rightarrow \bar{Q}_t(x,y) = 0$$

Therefore, we learn that $\bar{Q}_t$ fulfils the desired condition and fulfils the KFE. Therefore, we have proved that we can swap out $Q_t$ for $\bar{Q}_t$ and we will have an CTMC with the same marginals. $\square$

Now, let us return to the proof of Proposition 8.1. Given a rate matrix $Q_t$, we can now use a one-way rate matrix $\bar{Q}_t$ with the same marginals and define function:

$$F_t(\tau, i|x) = \begin{cases} \bar{Q}_t(y, x) & \text{if } Q_t(y, x) > 0 \\ -\bar{Q}_t(x, y) & \text{otherwise} \end{cases} \quad \text{where } y = \text{Swap}(x, i, \tau)$$

By construction, it holds that $F_t(\tau, i|x)$ is locally equivariant and that $[F_t(\tau, i|x)]_+ = \bar{Q}_t(y, x)$. This finishes the proof.

# E. Local equivariance of ConvNet

To verify the local equivariance, one can compute

$$\begin{aligned} G_t^\theta(\tau, i|x) &= (P_t^\theta(e_\tau) - P_t^\theta(x_i))^T H_t^\theta(i|x) \\ &= -(P_t^\theta(x_i) - P_t^\theta(e_\tau))^T H_t^\theta(i|x) \\ &= -(P_t^\theta(x_i) - P_t^\theta(e_\tau))^T H_t^\theta(i|\text{Swap}(x, i, \tau)) \\ &= -G_t(x^i, i|\text{Swap}(x, i, \tau)), \end{aligned}$$

where we have used the invariance of the projection head $H_t^\theta(i|x)$ to changes in the $i$-th dimension. This shows the local equivariance.

## E.1. Universal representation of decomposition into locally-invariant prediction head

In this section, we answer the question: To what extend do we reduce the expressive power of the neural network by imposing the above conditions? To show a universal representation theorem, define the polynomial basis features

$$P(x) = (1, x^1, x^2, x^3, x^4, \cdots)$$

i.e. we assume infinite width of the neural network output here.

**Proposition E.1.** *Any continuous function $\tilde{F} : C \to \mathbb{R}^d$ defined on a compact set $C \subset \mathbb{R}^d$ be approximated as follows: For any $\epsilon > 0$, there is a function $F = (F_1, \cdots, F_d) : C \to \mathbb{R}^D$ that can be decomposed in*

$$F_i(x) = P(x)^T H_i(x)$$

*for a locally-invariant function $H : C \to (\mathbb{R}^k)^d$ (i.e. $H_i(x)$ is independent of $x_i$) for some $k$ such that*

$$\sup_{x \in C} \|\tilde{F}(x) - F(x)\| < \epsilon$$

*In other words, for infinite width $k$, this representation is universal.*

*Proof.* Further, let $\tilde{F} : C \to \mathbb{R}^d$ be a continuous function defined on a compact set $C$. Further, write $\tilde{F} = (\tilde{F}_1, \cdots, \tilde{F}_d)$ for the coordinate functions. Then fix a $j$. By the Stone-Weierstrass theorem, there is a polynomial $p^j(x_1, \ldots, x_d)$ of the form

$$p^j(x_1, \ldots, x_d) = \sum_{k_1, \ldots, k_d \geq 0} a_{k_1, \ldots, k_d}^j x_1^{k_1} \ldots x_d^{k_d}$$

where $a_{k_1, \ldots, k_d} \neq 0$ only for a finite number of coefficients such that

$$\sup_{x \in C} |\tilde{F}_j(x) - p^j(x)| < \frac{\epsilon}{d}$$

Then we can re-express that polynomial via

$$\begin{aligned} p^j(x_1, \ldots, x_d) &= \sum_{k_i \geq 0} x_i^{k_i} \left( \sum_{k_i \geq 0, i \neq j} a_{k_1, \ldots, k_d} \prod_{j \neq i} x_j^{k_j} \right) \\ &= \sum_{k_i \geq 0} P_{k_i}(x_i) H_{i, k_i}(x) \\ &= P(x_i)^T H_i(x) \\ &=: F_i(x) \end{aligned}$$

Note that $H = (H_1, \cdots, H_d)$ is locally invariant. Therefore, we can conclude that $F = (F_1, \cdots, F_d)$ fulfills the desired conditions:

$$\sup_{x \in C} \|\tilde{F}(x) - F(x)\| \le \sum_{i=1}^{d} \sup_{x \in C} |\tilde{F}_i(x) - F_i(x)| < \sum_{i} \frac{\epsilon}{d} = \epsilon$$

As $\epsilon > 0$ was arbitrary, this finishes the proof. $\qquad \square$

## F. Recovering loss functions for CTMC models via RNDs

We discuss here in more detail how the Radon-Nikodym derivatives (RNDs) presented in Proposition 5.1 relate to the construction of loss function for CTMC generative models, also called "discrete diffusion" models. The connection lies in the fact that the loss function of these models relies on RNDs of two CTMCs running both in forward time. We can prove the following statement:

**Proposition F.1.** *Let $\mu, \nu$ be two initial distributions over $S$. Let $Q_t, Q'_t$ be two rate matrices. Then the Radon-Nikodym derivative of the corresponding path distributions in forward time over the interval $[0, t]$ is given by:*

$$\log \frac{\mathrm{d}\overrightarrow{\mathbb{P}}^{\mu,Q}}{\mathrm{d}\overrightarrow{\mathbb{P}}^{\nu,Q'}}(\mathbf{X})$$

$$= \log \frac{\mathrm{d}\mu}{\mathrm{d}\nu}(X_0) + \int_0^t Q_s(X_s, X_s) - Q'_s(X_s, X_s) ds$$

$$+ \sum_{s, X_s^- \ne X_s} \log \left( \frac{Q_s(X_s, X_s^-)}{Q'_s(X_s, X_s^-)} \right)$$

*where we sum over all points where $X_s$ jumps in the last term.*

The proof of the above formula is very similar to the proof of Proposition 5.1 and an analogous formula also appeared in (Campbell et al., 2024), for example. The above proposition allows us to by compute the KL-divergence:

$$D_{KL}(\overrightarrow{\mathbb{P}}_1^{\mu,Q} || \overrightarrow{\mathbb{P}}_1^{\nu,Q'})$$

$$\le D_{KL}(\overrightarrow{\mathbb{P}}^{\mu,Q} || \overrightarrow{\mathbb{P}}^{\nu,Q'})$$

$$= \mathbb{E}_{\mathbf{X} \sim \overrightarrow{\mathbb{P}}^{\mu,Q}} \left[ \log \frac{\mathrm{d}\overrightarrow{\mathbb{P}}^{\mu,Q}}{\mathrm{d}\overrightarrow{\mathbb{P}}^{\nu,Q'}}(\mathbf{X}) \right]$$

$$= D_{KL}(\mu || \nu) + \mathbb{E}_{\mathbf{X} \sim \overrightarrow{\mathbb{P}}^{\mu,Q}} \left[ \int_0^1 Q_t(X_t, X_t) - Q'_t(X_t, X_t) dt + \sum_{t, X_t^- \ne X_t} \log \left( \frac{Q_t(X_t, X_t^-)}{Q'_t(X_t, X_t^-)} \right) \right]$$

$$= D_{KL}(\mu || \nu) + \mathbb{E}_{t \sim \mathrm{Unif}_{[0,1]}, x_t \sim \overrightarrow{\mathbb{P}}_t^{\mu,Q}} [Q_t(X_t, X_t) - Q'_t(X_t, X_t)]$$

$$+ \mathbb{E}_{\mathbf{X} \sim \overrightarrow{\mathbb{P}}^{\mu,Q}} \left[ \sum_{t, X_t^- \ne X_t} \log \left( \frac{Q_t(X_t, X_t^-)}{Q'_t(X_t, X_t^-)} \right) \right]$$

$$= D_{KL}(\mu || \nu) + \mathbb{E}_{t \sim \mathrm{Unif}_{[0,1]}, x_t \sim \overrightarrow{\mathbb{P}}_t^{\mu,Q}} [Q_t(X_t, X_t) - Q'_t(X_t, X_t)]$$

$$+ \int_0^1 \mathbb{E}_{X_t \sim \overrightarrow{\mathbb{P}}_t^{\mu,Q}} \left[ \sum_{y \ne X_t} Q_t(y; X_t) \log \left( \frac{Q_t(y; X_t)}{Q'_t(y, X_t)} \right) \right] \mathrm{d}t$$

$$= D_{KL}(\mu || \nu) + \mathbb{E}_{t \sim \mathrm{Unif}_{[0,1]}, X_t \sim \overrightarrow{\mathbb{P}}_t^{\mu,Q}} \left[ \sum_{y \ne X_t} Q'_t(y, X_t) - Q_t(y, X_t) + Q_t(y; X_t) \log \left( \frac{Q_t(y; X_t)}{Q'_t(y, X_t)} \right) \right]$$

where we have used the data processing inequality in the first term. Having a parameterized model $Q_t' = Q_t^\theta$, this leads to the following loss:

$$L(\theta) = D_{KL}\left(\overrightarrow{\mathbb{P}}^{\mu,Q} || \overrightarrow{\mathbb{P}}^{\mu,Q_t^\theta}\right)$$

$$= D_{KL}(\mu||\nu) + \mathbb{E}_{t\sim\text{Unif}_{[0,1]},X_t\sim\overrightarrow{\mathbb{P}}_t^{\mu,Q}}\left[\sum_{y\neq X_t} Q_t^\theta(y, X_t) - Q_t(y, X_t)\log\left(Q_t^\theta(y, X_t)\right)\right] + C$$

where $Q_t$ is some reference process. The above recovers loss functions in the context of CTMC and jump generative models (Campbell et al., 2022; Gat et al., 2025; Shaul et al., 2024; Campbell et al., 2024) and Euclidean jump models (Holderrieth et al., 2024, Section D.1.). Note that the above loss cannot be used for the purposes of sampling in a straight-forward manner because we do not have access to samples from the marginals of the ground reference $\overrightarrow{\mathbb{P}}^{\mu,Q}$.

# G. Numerical experiments

All code for experiments can be found under the following link: https://github.com/malbergo/leaps/.

We briefly explain the Ising model here. The Potts model is similar and we refer to details for the Potts model to the DISCS benchmark (Goshvadi et al., 2023). Configurations of the $L \times L$ Ising lattice follow the target distribution $\rho_1(x) = e^{-\beta H(x) + F_1}$ where $\beta$ is the inverse temperature of the system, $F_1$ the free energy, and $H(x) : \{-1, 1\}^{L \times L} \to \mathbb{R}$ is the Hamiltonian for the model defined as

$$H(x) = -J\sum_{\langle i,j \rangle} x_i x_j + \mu \sum_i x_i. \tag{21}$$

Here, $J$ is the interaction strength, $\langle i, j \rangle$ denotes summation over nearest neighbors of spins $x_i, x_j$ and $\mu$ is the magnetic moment. Neighboring spins are uncorrelated at high temperature but reach a critical correlation when the temperature drops below a certain threshold. As observables, we use the histograms of the magnetization $M(x) = \sum_i x_i$ compared to ground truth, as well as the two point connected correlation function

$$G_{\text{conn}}(r) = \mathbb{E}[x_i x_{i+r}] - \mathbb{E}[x_i]\,\mathbb{E}[x_{i+r}]. \tag{22}$$

The latter is a measure of the dependency between spin values at a distance $r$ in lattice separation.

For evaluation, we use the self-normalized definition of the effective sample size such that, given the log weights $A_t$ associated to $N$ CTMC instances, the ESS at time $t$ in the generation is given by:

$$\text{ESS}_t = \frac{\left(N^{-1}\sum_{t=1}^N \exp\left(A_t^i\right)\right)^2}{N^{-1}\sum_{i=1}^N \exp\left(2A_i^i\right)} \tag{23}$$

