# OpenReview forum: "LEAPS: A discrete neural sampler via locally equivariant networks"
_ICML.cc/2025/Conference — ICML 2025 poster_

### Official Review · Reviewer_8trc · 2025-02-19

**Overall Recommendation:** 2

**Summary:**

This paper introduces a continuous-time diffusion sampler that operates along an annealed energy path in the discrete domain. The approach is trained using a PINN-based objective as an upper bound for the Log Variance Loss. The authors propose locally invariant neural network architectures for parametrization. The method is experimentally validated on a 15x15 Ising Model.

**Claims And Evidence:**

While the approach is theoretically sound, the empirical evidence is limited. The study includes only one experiment on the 15x15 Ising Model and lacks ablation studies for the proposed methods. Additionally, crucial details for reproducing the experiments are missing. Although a locally invariant MLP architecture is proposed, no experiments using this architecture are presented.

**Essential References Not Discussed:**

- [1] and [2] should also be cited in this context. [1] is seminal work in discrete samplers and [2] should be cited together with Nicoli et al., 2020.
- [3] and [4] are highly relevant discrete diffusion sampler papers. The Annealed Noise Distribution in [1] resembles a discrete variant of the transport proposed in this paper. [4] applies diffusion samplers in the discrete domain in statistical physics.
- [5] also proposes a PINN-based loss to learn a transport between a prior and target distribution in the continuous domain

**References:**

1. Wu, Dian, Lei Wang, and Pan Zhang. "Solving statistical mechanics using variational autoregressive networks." Physical review letters 122.8 (2019): 080602.
2. McNaughton, B., et al. "Boosting Monte Carlo simulations of spin glasses using autoregressive neural networks." Physical Review E 101.5 (2020): 053312.
3. Sanokowski, S., Hochreiter, S., & Lehner, S. A Diffusion Model Framework for Unsupervised Neural Combinatorial Optimization. In *Forty-first International Conference on Machine Learning*.
4. Sanokowski, Sebastian, et al. "Scalable Discrete Diffusion Samplers: Combinatorial Optimization and Statistical Physics." arXiv preprint arXiv:2502.08696 (2025).
5. Tian, Yifeng, Nishant Panda, and Yen Ting Lin. "Liouville Flow Importance Sampler." *arXiv preprint arXiv:2405.06672* (2024).

**Experimental Designs Or Analyses:**

The experiments are not reproducible due to several reasons:
- No information is provided about the hyperparameter selection process or the final hyperparameters used, such as learning rate and the number of samples during training or evaluation.
- It is unclear whether the Ising Model is defined on a periodic or non-periodic grid.
- The value of $\mu$ in Eq. 18 is not specified, and there is no comparison to theoretically available solutions for the Free Energy, Entropy, and Internal Energy when $\mu = 0$.

**Methods And Evaluation Criteria:**

The model is evaluated based on effective sample size, two-point correlation function, and magnetization.
This makes sense, however, an estimation based on log Z, ELBO, internal Energy and entropy could be added, especially when $\mu = 0$, where these values are theoretically available.

**Other Comments Or Suggestions:**

This paper has the potential to be very impactful, but it requires more experimental evaluation and detailed descriptions of the experiments.
Experiments on Spinn Glasses as in [4] would also be nice to have.

**Other Strengths And Weaknesses:**

**Strengths:**
- The proposed framework and locally invariant architectures are interesting.

**Weaknesses:**
- Many ablation studies are missing, such as comparisons between LV loss vs. PINN loss and locally invariant networks vs. vanilla architectures.
- The experiments are scarce and not reproducible with the available information.
- There is no comparison to other discrete domain diffusion samplers or autoregressive samplers.

**Questions For Authors:**

- How does a non-locally invariant architecture compare to the proposed architectures?
- How was the proposal distribution $\mathbb{Q}$ chosen?
- Why were $\beta = 0.7$ and $J = 0.4$ specifically chosen? The correlation length seems quite small for this choice (see Fig. 4 right). This might have the consequence that the problem is not hard for this choice of parameters.
- Why is there no comparison on the ising model, with $\mu = 0$, $J = 1$ and $\beta = 0.4407$ at the critical temperature where the correlation length does not decay and theoretical solutions can be used as a baseline?

**Relation To Broader Scientific Literature:**

The paper provides a broader relation to scientific literature but omits some relevant papers, particularly those related to discrete domain diffusion samplers.

**Theoretical Claims:**

Theoretical claims were partially checked.

---

> ### Author Rebuttal · Authors · 2025-04-01
>
> Thank you for your thorough and valuable feedback on our manuscript. Below we itemize and address your comments and concerns.
>
> **Further experiments and benchmarks:**
> - We have included additional experiments, as possible within the limited time frame. Specifically, we run additional tests of the Ising model at the critical parameters you suggested, and benchmarked this setup against the samplers that reviewer YU3Q suggested (DISCS) and annealed importance sampling (AIS). The results are provided in the anonymous google drive https://tinyurl.com/leapsicml where we see that the LEAPS performs favorably in this setting. To save space to address your other concerns, **please see our response to YU3Q for a thorough discussion of these results.**
>
> **No experiments with locally equiv MLP**
>
> - We have tested the MLP architecture and it has performed significantly worse. The convolutional architecture is geometrically translation equivariant, which is known in deep learning to outperform MLPs if the data (e.g. the statistical physics models we consider) is symmetric. We therefore do not consider this a noticable benchmark.
>
> **Experimental details.** Thank you for letting us know about missing details for replication. Here we supply an exhaustive list:
> - Learning rate=5e-3
> - Hyperparameter selection: We peform manual optimization of the neural network architectures and hyperparameters.
> - Batch size 256 walkers, stored in a replay buffer. New walkers are added to the buffer every 50 training steps
> - Training iterations: 30k
> - Ising model: periodic lattice.
> - External magnetic field: $\mu=0$.
> - Comparison to theoretically available solutions: The theoretically available solutions are infinite grid sizes. We run Glauber dynamics that simulate the actual physics of the system for a sufficiently long time to recover the statistics that we compare against.
>
> We will add the above details to the paper. Please let us know if you are missing other details.
>
> **Essential References Not Discussed:**
> - Thank you for pointing us to these citations. We will happily add them to the text when we are allowed to edit it. We note that the paper by Tian et al is cited already in multiple locations of the manuscript.
> - One distinguishing feature of our approach to autoregressive models is that we can draw samples with fewer than d steps where d is the dimension. For example, we achieved an ESS of 69% in $<100$ steps for the Ising model. An autoregressive could only achieve the same results in $d=L^2$ steps. Therefore, our effective ESS will be higher.
>
> **Non-locally invariant architecture clarification**
>
> - Thank you for bringing this up. As stated in section 7, a non-locally equivariant architecture would require $\mathcal{O}(N*d)$ evaluations of the network for computation for $N$ the number of neighbors and $d$ the dimension. This number is prohibitively large for any $d$ that is interesting for applications, e.g. $d=225$ in the Ising experiment. We do not consider a need to give experiments to show that this is unfeasible. Of course, one could *train* a neural network architecture to be locally equivariant via an additional loss but then lose the computation of *exact* importance weights, which is one of the main goals of our paper.
>
> **Log-variance.**
> - We thank the reviewer for bringing up the log-variance loss. Evaluating the log-variance divergence is computationally much more expensive here, which is why we do not benchmark against it. Specifically, the log-variance divergence is $K$ times more expensive in memory, where $K$ is the number of simulation steps. The reason for that is as follows: The log-variance loss is not valid when sampled pointwise (it requires the whole trajectory). In contrast, the PINN loss can be evaluated pointwise. However, the PINN objective generally requires a “discrete divergence” computation (the $\mathcal{K}_t$ operator). We get rid of the cost of this discrete divergence with the locally equivariant networks. Note that this is not necessarily true in the continuous case, where the log-var can avoid the computation of a divergence via Ito-integrals.
>
> **Proposal distribution.**
> - We use the model itself as a proposal distribution.
>
> Finally, we like to emphasize that the we consider the the main contribution of this work the introduction of a new paradigm for learning to sampling from discrete distributions. We therefore stress our methdological and theoretical contributions:
> - A new derivation of Radon-Nikodym derivatives of path measures
> - Proactive importance sampling as an IS scheme for CTMCs
> - A PINN-objective for discrete samplers by bounding the variance of the IS weights
> - Local equivariance as a symmetric constraint to enable scalable proactive IS
> - Locally equivariant neural network architectures such as convolutional neural networks
>
> Thank you again for your valuable review. We hope that this addresses your questions. We would accordingly appreciate any increase in rating you see fit.

---

### Official Review · Reviewer_esu2 · 2025-03-13

**Overall Recommendation:** 4

**Summary:**

The goal of this paper is to draw samples from a distribution $\rho_1$, known up to a normalization constant, over a discrete space.

One way to do so is to simulate a prescribed path of marginal distributions ${(\rho_t)}_{t \in [0, 1]}$ that ends with the desired target.

To do so, the authors introduce a generic process ${(p_t)}_{t \in [0, 1]}$ driven by a Markov operator $Q_t$,

and then consider the reweighted process $(w_t p_t)_{t \in [0, 1]}$,

where the reweighting is done by the probability distribution $w_t(x) \propto \mathbb{E}_{A_t | X_t = x}[\exp(A_t)]$.

This reweighted sampling process simulates exactly the prescribed path of marginals, if either:

1. the Markov operator $Q_t$ verifies (Eq 7): $\partial_t \rho_t(x) = \sum_{y \in S} Q_t(x, y) \rho_t(y)$

2. the log-unnormalized weights $A_t$ verify (Eq 11): $\partial_t A_t = (1 / \rho_t(X_s)) \big( \partial_t \rho_t(X_s) - \sum_{y \in S} Q_t(X_s, y) \rho_t(y) \big)$

The authors propose to enforce the equation on $Q_t$ by minimizing its violation, which can be written as a PINN loss. In that loss, the authors propose an computationally efficient parameterization of $Q_t$ based on equivariant neural networks.

## update after rebuttal

I appreciate that additional experiments were added, where the authors compare against baseline methods that were lacking in the original submission. Sampling in discrete spaces is a rapidly evolving field with a broad literature, ranging from statistical physics to mainstream machine learning, so finding all the relevant literature is not so obvious. It seems like the authors agreed to cite the references that were rightly brought up by many reviewers. I agree with reviewer 8trc that further clarity would be welcome in the experiments (using other metrics than ESS) and the text (claims about auto-regressive models and the log-variance loss). Yet overall, the paper is already an interesting contribution to the field. So I will maintain my positive score.

**Claims And Evidence:**

Yes.

**Essential References Not Discussed:**

Related work is discussed.

**Experimental Designs Or Analyses:**

I looked at the reported results which are encouraging. I have a few questions, however:

- **What does "no transport" refer to?**

In Figure 4, what does "no transport" refer to? I read in the caption that it denotes the case "of using annealed dynamics with just the marginal preserving MCMC udpates to show that the transport from Q_t is essential". I don't understand what this corresponds to exactly in the text. Could the authors provide more detail?

- **More datasets**

The experiments on the Ising model, while principled (because the ground truth is known) and encouraging, are limited. Did the authors benchmark against other target distributions, such as quantized MNIST as in [1].

- **LEAPS uncorrected vs. LEAPS**

In Figure 1, the authors contrast LEAPS (perfect transport and reweighting) with uncorrected LEAPS (reweighting only). Which $Q_t$ is used for the uncorrected LEAPS? Because the distinction between LEAPS and uncorrected LEAPS is central to the authors' paper, it would be useful to see both these results in Figure 4 as well.


[1] Discrete Langevin Sampler via Wasserstein Gradient Flow. Haoran Sun, Hanjun Dai, Bo Dai, Haomin Zhou, Dale Schuurmans. 2022.

**Methods And Evaluation Criteria:**

The evaluation is encouraging.

**Other Comments Or Suggestions:**

-

**Other Strengths And Weaknesses:**

-

**Questions For Authors:**

1. **Confused about motivation of the equivariant parameterization of the rate matrix**

In section 7, the authors first write an equation involving $Q_t^{\theta}(y^i, i | x)$ and $Q_t^{\theta}(x^i, i | y)$. If I correctly understand, each of these terms should be evaluated for each neighbor $y$, so roughly 2 |N(x)| times.

After enforcing equivariance, the equation now only involves $F_t^{\theta}(y^i, i | x)$. As I understand it, this term should also be evaluated for each neighbor $y$ so |N(x)| times. What am I missing?

2. **Connection with NETS**

The authors mention that their paper can be understood as an extension of [1]. The comparison is

| NETS    | LEAPS |
| -------- | ------- |
| ALD  | CTMC |
| ALD + perfect additional transport (Eq 16) |  CTMC + perfect rate matrix  |
| ALD + reweighting (Eqs 10, 11) |  CTMC + reweighting |

One difference is that the vanilla ALD defines a specific, tractable Markov kernel, that does not require learning and simulates *approximately* the prescribed path.
In contrast, CTMC is defined by a general Markov kernel that does not necessarily simulate approximately the prescribed path.

Do the authors have an idea of how to choose a CTMC that would be a closer comparison to ALD? Specifically, do they have an idea of what would be a default, tractable choice of $Q_t$, that does not require learning and simulates *approximately* the prescribed path?

3. **Question about the PINN loss**

The PINN loss in Proposition 6.1. encourages $K_s \rho_s = \partial_s F_s$. Shouldn't it encourage $K_s \rho_s = \partial_s A_s$ from Eq 11.

4. **Interpolation schemes**

The authors define a general interpolation in Eq 2. In Figure 4, the authors use an interpolation that is specific to the Ising model. In Figure 1, which interpolation do the authors use? In the text after equation 3, the authors say that when the initial distribution is uniform, then we get $\rho_t \propto exp(-t U_1(x))$ but this is the case when the interpolation is a geometric mean. In discrete diffusion models, the interpolation is an arithmetic mean between the start and end distributions. Can the authors clarify which interpolations they use? Also, have the authors tried different interpolations or have any thoughts on how that choice affects the learning of $Q_t$?

[1] NETS: A Non-Equilibrium Transport Sampler. Michael S. Albergo, Eric Vanden-Eijnden. 2024.

**Relation To Broader Scientific Literature:**

This paper relates to the sampling literature.

**Theoretical Claims:**

Not in detail but they seem coherent with previous literature.

---

> ### Author Rebuttal · Authors · 2025-04-01
>
> We thank the reviewer for their valuable feedback on our work. Below, we provide answers to your questions and concerns.
>
> **Additional benchmarks and experiments.** We obtained new experimental results. The results can be found under this anonymous link, in which we show that our method performs well at the critical phase of the Ising model and compares favorably against existing samplers (see more below): https://tinyurl.com/leapsicml
> We use two benchmarks:
> 1. **LEAPS vs. MCMC.** We benchmark LEAPS against a diversity of MCMC samplers on the critical temperature of the Ising model. The results can be found in figure 1 in the google drive folder. As one can see, LEAPS achieves an effective sample size (ESS) of $\sim 70$% vs ESS$<0.1$% of MCMC-based samplers.
> 2. **LEAPS vs. AIS:** We also benchmark LEAPS against annealed importance sampling (AIS), see figure 3. As one can see, even with $100k$ simulation steps, AIS only achieves an ESS of $<30$% vs an ESS of $69$% of LEAPS with 100 simulation steps.
> 3.
> Please see our response to YU3Q for a thorough discussion of these results.
>
> **Questions.** We address your remaining questions and comments in the following:
> - ```What does "no transport" refer to?```: This refers to not using the neural network at all. We show via this ablation that it is truly the neural network (and not the AIS) that enables the sampling from the distribution with only a few samples. We ran additional benchmarks on this that we explained above (AIS vs LEAPS) not using AIS at all for LEAPS.
> - ``LEAPS uncorrected vs. LEAPS``: For the simulation of the continuous-time Markov chain, we always use the rate matrix $Q_t$ represented by our network. Due to the local equivariance, we get importance weights for free. "LEAPS uncorrected" means that we do not use the importance weights (at perfect training, this would not be necessary). "LEAPS corrected" means that we use the importance weights to reweight samples towards the correct distribution. Both parts are essentials as our ablations show.
> - ```More datasets and benchmarks:``` We provided other benchmarks, in particular against a series of MCMC samplers. We further actively run experiments on the Potts model, another model from statistical physics. The only other established benchmark for discrete sampling is combinatorial optimization. However, we note that metrics used for combinatorial optimization do not aim to faithfully sample from a distribution or any observable thereof, where importance weights could be used. Rather it tries to find low energy states. This puts our method fundamentally at a disadvantage.
> - ```Clarifying equivariance```: We note that for an input $x$ the neural network outputs $F_t^\theta(y^i,i|x)$ for every $y^i,i$ for token $y^i$ and index $i$ (this is also true for discrete diffusion models, it returns a rate per neighbor). Therefore, this requires only one forward pass. Thank you for bringing it up that this needs further clarification. We will highlight the distinction between input and output of the neural network more in future versions.
> - ```Connection with NETS/ALD```: Here is the analogy. Because ALD is like running Langevin dynamics locally on some $\rho_t \sim e^{-U_t}$ for fixed $t$ and then iterating $t \rightarrow t + dt$, so too can we add any local MCMC kernel to the dynamics of the CTMC that preserves detailed balance. This would "simulate approximately the prescribed path" just like Langevin does in the continuous setting. Does that make sense? We will add a remark about this in the text.
> - ```Question about the PINN loss```: The goal of LEAPS (and of all importance sampling schemes) is that the weights have as low variance as possible. In our case, this would mean that $A_t$ is just a constant in time and space. In other words, this means $\partial_sA_s=0$. As we show in the paper, this is equivalent to $\mathcal{K}_s\rho_s=\partial_sF_s$. Thank you for bringing this up. We will highlight this in future versions of our work.
> - ```Interpolation schemes```: The interpolation scheme is given by a time-dependent Ising model for coupling constant $J_t=tJ_1$ where here $J_1=1$. As you point out, this is equivalent to temperature annealing, i.e. $\rho_t(x)\propto \exp(-tU_1(x))$. As also pointed out, discrete diffusion models use arithmetic means (or probabilistic mixtures) between a uniform (or mask) and the target distribution. In principle, one could use such an interpolation. We focuses on temperature annealing because it is common in the sampling literature and it allows to sample from the Ising model for higher temperatures on the fly (by simply stopping at earlier time points). Therefore, this is a natural and physical way. We have explored various annealing speeds and schedules (i.e. time parameterizations) for the annealing that is happening during training and found the schedule of $t\to \sqrt{t}$ to perform best.

---

### Official Review · Reviewer_YMxM · 2025-03-14

**Overall Recommendation:** 4

**Summary:**

The paper proposes locally equivariant functions, a compact neural parameterization of rate matrices in continuous-time Markov processes over discrete state spaces.  This effectively allows them to use recent "discrete diffusion" models as proposals in annealed importance sampling and sequential Monte Carlo over discrete spaces.  They train these via a variational upper bound on the log-variance divergence. The experiment with a 2D Ising model that has a computable ground truth shows an improvement in performance using the resulting proposals over proposals designed without any measure-transport or diffusion machinery.

**Claims And Evidence:**

Yes.

**Essential References Not Discussed:**

I could not tell the authors if they were missing a discrete diffusion paper.

**Experimental Designs Or Analyses:**

There was only one experiment, and its design looks fine.  In the course of reviewer discussion, the authors have added additional benchmarks and experiments, which appear quite thorough.

**Methods And Evaluation Criteria:**

Yes.

**Other Comments Or Suggestions:**

N/A

**Other Strengths And Weaknesses:**

The paper's main weakness is that it gives exactly one experiment in which the only baseline against something other than a locally-equivariant sampler is "no transport".  Could methods from other works cited not be compared to the present method?

After review the paper has become significantly stronger through the addition of more experiments.

**Questions For Authors:**

N/A

**Relation To Broader Scientific Literature:**

The paper contextualizes itself adequately within the literature.

**Theoretical Claims:**

Proofs are in the appendix and were not checked, but the theorems and propositions themselves do look sensible to me.

---

> ### Author Rebuttal · Authors · 2025-04-01
>
> Thank you for your valuable feedback on our work. Below we address your questions and provide new information on additional experiments.
>
> **Updates to experiments: harder sampling problem and comparison benchmarks**
> We obtained new experimental results and benchmarks as possible within the limited time frame. The results can be found at this anonymous link, in which we show that our method performs well at the critical point of the Ising model and compare it extensively against other methods in this hard regime (see more below): https://tinyurl.com/leapsicml
>
>
> **Additional benchmarks (1) - LEAPS vs. MCMC.** We use the DISCS benchmark [1] for discrete samplers to compare LEAPS to various MCMC samplers. We also adapt the parameters of the Ising model to the critical temperature for $\beta=0.4407$. Note that this makes the problem harder (both for traditional methods and for LEAPS). The results can be found in figure 1 in the google drive folder. As one can see, LEAPS achieves an effective sample size (ESS) of $\sim 70$% vs an ESS of $<0.1$% of MCMC-based samplers. This shows how LEAPS effectively converts a simple distribution into a highly complex distribution that can only be sampled with many MCMC runs. In figure 3, we also show that the samples recover known physical observables of the Ising model. Finally, we note that there are significant difficulties in creating a fair assessment between non-equilibrium (LEAPS) and equilibrium methods (MCMC):
> - The ESS is measured differently  in DISCS because it focuses on equilibrium MCMC methods, rather than non-equilibrium sampling  methods such as LEAPS.
> - The ESS quantities for non-equilibrium methods such as LEAPS do not take into account the number of steps used for annealing (i.e. for simulation of the continuous-time Markov chain). To account for this, we have also plotted the same quantities normalized by the number of function evaluations in Figure 1 (ESS/NFEs). However, LEAPS does *not* require evaluations of the energy at inference time but only of the neural network, while MCMC methods require energy evaluations (see metrics used in DISCS [1]). The best way we account for that now would be to normalize by the NFEs as measured in neural net evaluations. However, note that this does not measure efficiency or wall clock time, which would highly dependent on implementation and is usually not measured in the literature on neural sampling methods. The main goal of LEAPS is to solve the sampling problem, not efficiency.
>
> We hope that our new experiments showcase the efficacy of the LEAPS algorithm, while also explaining the difficulty in benchmarking LEAPS vs traditional equilibrium methods.
>
> **Additional benchmarks (2) - LEAPS vs. AIS.** We also benchmark LEAPS against annealed importance sampling (AIS) [2]. AIS is a standard non-equilibrium sampling technique. Note that here, ESS is measured in the same way and the two methods are exactly comparable. In Figure 2, we plot the number of AIS steps vs the ESS of AIS. As one can see, even with $100k$ simulation steps, AIS only achieves an ESS of $<30$% vs an ESS of $69$% of LEAPS with 100 simulation steps. We note that LEAPS is a true extension of AIS and we can run AIS with the learned transport without changing the weights. In addition it can be turned into a sequential monte carlo sampler by performing resampling along the trajectory. However, here, for the purposes of benchmarking we turn off the AIS (here and above) to showcase that the learnt transport via the neural network enables the efficiency boost.
>
> **Comparison to other works cited:** Many of the works that we cite in our work are about the mathematical realization of discrete diffusion models and sampling algorithms for continuous variables. To the best of our knowledge, we do not know of another paper which learns a neural sampler for discrete distributions via Continuous Time Markov Chain. Therefore, we cannot benchmark against them. If you know of any, please let us know.
>
> Finally, we would like to emphasize that the we consider the the main contribution of this work the introduction of a new paradigm for learning to sampling from discrete distributions. We therefore emphasize again our methdological and theoretical contributions:
> - A new derivation of Radon-Nikodym derivations of path measures
> - Proactive importance sampling as an IS scheme for CTMCs
> - A PINN-objective for discrete samplers by bounding the variance of the IS weights
> - Local equivariance as a symmetric constraint to enable scalable proactive IS without losing representational power
> - Locally equivariant neural network architectures such as LE-convolutional neural networks
>
> Thanks again for your valuable feedback. If these results have fulfilled your questions and concerns, we would greatly appreciate any increase in score.
>
> [1]. Goshvadi K, Sun H, Liu X, et al. DISCS: a benchmark for
> discrete sampling[J]. Advances in Neural Information Processing Systems, 2023, 36: 79035-79066.

---

### Official Review · Reviewer_YU3Q · 2025-03-24

**Overall Recommendation:** 4

**Summary:**

The authors propose an algorithm for sampling from discrete distributions by combining importance sampling with a learned continuous-time Markov chain (CTMC). They derive the importance weights via a Radon–Nikodym derivative for CTMCs and introduce locally equivariant neural architectures to ensure tractable learning of the rate matrices. Empirical results on a 2D Ising model demonstrate the sampling efficiency of the algorithm.

**Claims And Evidence:**

Most of the theoretical claims in the paper are supported by clear proofs or sound intuitions. However, while the empirical results on the 2D Ising model are promising, further evidence on additional benchmarks or experiments could strengthen the overall support for the method’s general applicability. (Please see "Experimental Designs Or Analyses" part)

**Essential References Not Discussed:**

It would be a good idea if the authors could discuss the family of newly proposed discrete sampling methods mentioned in [1], like Locally Balanced [2], Gibbs with Gradients [3],  Path Auxiliary
Sampler [4], Discrete Metropolis Adjusted Langevin Algorithm [5],
Discrete Langevin Monte Carlo [6].




References:


[1]. Goshvadi K, Sun H, Liu X, et al. DISCS: a benchmark for discrete sampling[J]. Advances in Neural Information Processing Systems, 2023, 36: 79035-79066.

[2]. Zanella G. Informed proposals for local MCMC in discrete spaces[J]. Journal of the American Statistical Association, 2020, 115(530): 852-865.

[3]. Grathwohl W, Swersky K, Hashemi M, et al. Oops i took a gradient: Scalable sampling for discrete distributions[C]//International Conference on Machine Learning. PMLR, 2021: 3831-3841.

[4]. Sun H, Dai H, Xia W, et al. Path auxiliary proposal for MCMC in discrete space[C]//International Conference on Learning Representations. 2021.

[5]. Zhang R, Liu X, Liu Q. A Langevin-like sampler for discrete distributions[C]//International Conference on Machine Learning. PMLR, 2022: 26375-26396.

[6]. Sun H, Dai H, Dai B, et al. Discrete langevin samplers via wasserstein gradient flow[C]//International Conference on Artificial Intelligence and Statistics. PMLR, 2023: 6290-6313.

**Experimental Designs Or Analyses:**

There are two main concerns regarding the experiment part of this paper:


1. Insufficient benchmarks: Considering the many effective discrete sampling algorithms proposed in recent years, the paper would be more convincing if the authors compared their method against these state-of-the-art techniques. For example, they could refer to [1], which presents a set of advanced discrete samplers and provides useful comparisons between these samplers.


2. Lack of more experiments: The evaluation is limited to the Ising model. To more comprehensively demonstrate the algorithm’s effectiveness, the authors should consider additional experiments on other tasks, such as other classical graphical models, combinatorial optimization problems, or generative tasks. The settings and benchmarks in [1] could serve as a useful guide for expanding the experimental section.


References:


[1]. Goshvadi K, Sun H, Liu X, et al. DISCS: a benchmark for discrete sampling[J]. Advances in Neural Information Processing Systems, 2023, 36: 79035-79066.

**Methods And Evaluation Criteria:**

The proposed methods are well-suited to the problem at hand. The derivation of importance sampling weights via the Radon–Nikodym derivative is both theoretically sound and intuitively reasonable, and the parametrization of the rate matrix aligns with common practices in discrete Markov models. Moreover, the use of effective sample size (ESS) as the evaluation criterion in the 2D Ising model experiment is standard for assessing sampling efficiency, providing a clear metric for evaluating the algorithm’s performance.

**Other Comments Or Suggestions:**

There are several typos and unclear notations that need to be corrected:

1. **Line 110 (right):**
   "As we do not the normalization constant" should be "As we do not know the normalization constant."

2. **Equation 12 (second line):**
   It appears that $\mathbf{Y}$ might be a typo;

3. **Proposition 6.1:**
   The sampling notation should likely be $s \sim \mathrm{Unif}[0,t]$ instead of $s \sim \mathrm{Unif}[0,1]$.

4. **Equation 16 (second line):**
   The notation $y^j$ should be corrected to $y_j$.

**Other Strengths And Weaknesses:**

Strengths:

1. The paper is highly motivated, contributing to the challenging and important topic of discrete sampling, which has significant implications in statistics and machine learning.

2. The derivation of the proposed algorithm is intuitive and theoretically sound, and the approach is novel to me.


Weaknesses:

My main concern lies with the experimental section. Expanding the benchmarks and providing more extensive empirical comparisons, as discussed in the “Experimental Designs or Analyses” section, would strengthen the paper. With improvements to the experimental evaluation, the paper would make a stronger contribution.

**Questions For Authors:**

Please see "Experimental Designs Or Analyses" and "Other Comments Or Suggestions"

**Relation To Broader Scientific Literature:**

The paper provides its contributions to discrete sampling problems. In particular, it builds on prior advances in annealed importance sampling, sequential Monte Carlo methods, and recent discrete diffusion models that employ CTMCs. The introduction of locally equivariant neural architectures for parameterizing the rate matrix extends existing ideas from neural parameterizations in discrete Markov models.

**Theoretical Claims:**

I checked the proof of Proposition 5.1 and Theorem 5.2 roughly. Both proofs appear to be logically structured and correct.

---

> ### Author Rebuttal · Authors · 2025-04-01
>
> We thank the reviewer for their valuable feedback on our work. We are glad to read that the reviewer considers our work a "sound" and "novel" contribution. Below, we provide answers to your questions and concerns.
>
> We understand that the reviewer sees our experiments as the main area of improvement. Addressing this, we obtained new experimental results - as possible within the limited time frame. The results can be found under this anonymous link, in which we show that our method performs well at the critical phase of the Ising model and compares favorably against existing samplers (see more below): https://tinyurl.com/leapsicml
>
> We would also like to emphasize that the we consider the main contribution of this work the introduction of a new paradigm for learning to sampling from discrete distributions. We therefore stress our methodological and theoretical contributions:
> - A new derivation of Radon-Nikodym derivatives of path measures
> - Proactive importance sampling as an IS scheme for CTMCs
> - A PINN-objective for discrete samplers by bounding the variance of the IS weights
> - Local equivariance as a symmetric constraint to enable scalable proactive IS
> - Locally equivariant neural network architectures such as convolutional neural networks
>
> **Additional benchmarks (1) - LEAPS vs. MCMC.** Thank you for sharing the DISCS benchmark [1] for discrete samplers with us. To use this benchmark, we adapt the parameters of the Ising model to match theirs, i.e. we use the parameters for the critical temperature for $\beta=0.4407$. Note that this makes the problem harder (both for traditional methods and for LEAPS). We then ran benchmarks of LEAPS vs other discrete samplers (including the ones in the DISCS benchmark). The results can be found in figure 1 in the google drive folder. As one can see, LEAPS achieves an effective sample size (ESS) of $\sim 70$% vs ESS$<0.1$% of MCMC-based samplers. This shows how LEAPS effectively converts a simple distribution into a highly complex distribution that can only be sampled with many MCMC runs. In figure 3, we also show that the samples recover known physical observables of the Ising model. Finally, we note that there are significant difficulties in creating a fair assessment between non-equilibrium (LEAPS) and equilibrium methods (MCMC):
> - The ESS is measured differently  in DISCS because it focuses on equilibrium MCMC methods, rather than non-equilibrium sampling  methods such as LEAPS.
> - The ESS quantities for non-equilibrium methods such as LEAPS do not take into account the number of steps used for annealing (i.e. for simulation of the continuous-time Markov chain). To account for this, we also plot the same quantities normalized by the number of function evaluations in Figure 1 (ESS/NFEs). However, LEAPS does *not* require evaluations of the energy at inference time but only of the neural network, while MCMC methods require energy evaluations (see metrics used in DISCS [1]). The best way we account for that now would be to normalize by the NFEs as measured in neural net evaluations. However, note that this does not measure efficiency or wall clock time. The main goal of LEAPS is to solve the sampling problem, not efficiency.
>
> We hope that our new experiments showcase the efficacy of the LEAPS algorithm, while also explaining the difficulty in benchmarking.
>
> **Additional benchmarks (2) - LEAPS vs. AIS.** We also benchmark LEAPS against annealed importance sampling (AIS). AIS is a standard non-equilibrium sampling technique. Note that here, ESS is measured in the same way. In Figure 2, we plot the number of AIS steps vs the ESS of AIS. As one can see, even with $100k$ simulation steps, AIS only achieves an ESS of $<30$% vs an ESS of $69$% of LEAPS with 100 simulation steps. We note that LEAPS is a true extension of AIS (as well as SMC) and we can run AIS with the learned transport without changing the weights. However, for the purposes of benchmarking we turn off the AIS (here and above) to show that the learnt transport via the neural network is what enables the performance boost.
>
> **Other benchmarks.** Thank you for bringing up other possible benchmarks. We note that combinatorial optimization does not aim to faithfully recover a distribution or any observable thereof, where importance weights could be used. This puts our method fundamentally at a disadvantage. Please let us know if there are other benchmarks that come to mind that we can run to convince you of the potential of the method.
>
> **References.** Thank you for highlighting the references that we included in the updated draft of our work. We have benchmarked against the suggested methods (see above).
>
> We thank the reviewer again for the insightful response. If these results have addressed your concerns, we would greatly appreciate any increase in score.
>
> [1]. Goshvadi K, Sun H, Liu X, et al. DISCS: a benchmark for
> discrete sampling[J]. Advances in Neural Information Processing Systems, 2023, 36: 79035-79066.

---

### Decision · Program_Chairs · 2025-05-01

**Decision:**

Accept (poster)

**Comment:**

This paper proposes LEAPS, a novel discrete sampler that uses locally equivariant neural network architectures to parameterize the rate matrix of a continuous-time Markov chain (CTMC). The method is designed for efficient sampling from unnormalized discrete distributions. It integrates ideas from annealed importance sampling, non-equilibrium transport, and Radon–Nikodym derivatives over path measures. The experiments are conducted on a 2D Ising model.

All reviewers found the core idea compelling, particularly the use of importance sampling and locally equivariant networks within the CTMC framework. However, several reviewers noted that the empirical evaluation, limited initially to the 2D Ising model without any baselines, was too narrow. In response, the authors provided ESS comparisons on both the Ising and Potts models with several existing discrete samplers. While this significantly improves empirical support, relying solely on ESS is insufficient to demonstrate sampling performance. Moreover, the paper lacks ablation studies to isolate the contributions of each proposed component.

Overall, the paper presents a theoretically sound and promising approach to discrete sampling using locally equivariant neural networks. The methodology represents a valuable contribution to the field. However, due to the limited empirical evaluation, it remains difficult to assess the true impact of the proposed method. The authors are strongly encouraged to expand the empirical evaluation by including more comprehensive comparisons on different tasks, particularly on real-world datasets, and against a broader set of discrete sampling baselines.